# Phase separation of *Arabidopsis* EMB1579 controls transcription, mRNA splicing, and development

**Yiling Zhang**[1☯], **Zhankun Li**[1☯], **Naizhi Chen**[2☯], **Yao Huang**[1], **Shanjin Huang**[1]*

**1** Center for Plant Biology, School of Life Sciences, Tsinghua University, Beijing, China, **2** Key Laboratory of Plant Resources, Institute of Botany, Chinese Academy of Sciences, Beijing, China

☯ These authors contributed equally to this work.
* sjhuang@tsinghua.edu.cn

**Data Availability Statement:** All relevant data are within the paper and its Supporting Information files. RNA-seq data have been deposited at BioProject (https://www.ncbi.nlm.nih.gov/

## Abstract

Tight regulation of gene transcription and mRNA splicing is essential for plant growth and development. Here we demonstrate that a plant-specific protein, EMBRYO DEFECTIVE 1579 (EMB1579), controls multiple growth and developmental processes in *Arabidopsis*. We demonstrate that EMB1579 forms liquid-like condensates both in vitro and in vivo, and the formation of normal-sized EMB1579 condensates is crucial for its cellular functions. We found that some chromosomal and RNA-related proteins interact with EMB1579 compartments, and loss of function of *EMB1579* affects global gene transcription and mRNA splicing. Using floral transition as a physiological process, we demonstrate that EMB1579 is involved in *FLOWERING LOCUS C* (*FLC*)-mediated repression of flowering. Interestingly, we found that EMB1579 physically interacts with a homologue of *Drosophila* nucleosome remodeling factor 55-kDa (p55) called MULTIPLE SUPPRESSOR OF IRA 4 (MSI4), which has been implicated in repressing the expression of *FLC* by forming a complex with DNA Damage Binding Protein 1 (DDB1) and Cullin 4 (CUL4). This complex, named CUL4-DDB1[MSI4], physically associates with a CURLY LEAF (CLF)-containing Polycomb Repressive Complex 2 (CLF-PRC2). We further demonstrate that EMB1579 interacts with CUL4 and DDB1, and EMB1579 condensates can recruit and condense MSI4 and DDB1. Furthermore, *emb1579* phenocopies *msi4* in terms of the level of H3K27 trimethylation on *FLC*. This allows us to propose that EMB1579 condensates recruit and condense CUL4-DDB1[MSI4] complex, which facilitates the interaction of CUL4-DDB1[MSI4] with CLF-PRC2 and promotes the role of CLF-PRC2 in establishing and/or maintaining the level of H3K27 trimethylation on *FLC*. Thus, we report a new mechanism for regulating plant gene transcription, mRNA splicing, and growth and development.

## Introduction

Plant growth and development are tightly regulated in response to many endogenous and environmental signals by genetic and cellular programs that determine plant form. As sessile organisms, plants need to efficiently organize various cellular events to cope with the ever-changing

bioproject/) under the accession number
PRJNA653772.

**Funding:** This work was supported by grants from
the National Natural Science Foundation of China
(31471266 and 31421001). The research in the
Huang Lab is also supported by the funding from
Beijing Advanced Innovation Center for Structural
Biology. The funders had no role in study design,
data collection and analysis, decision to publish, or
preparation of the manuscript.

**Competing interests:** The authors have declared
that no competing interests exist.

**Abbreviations:** A3SS, alternative 3′ splice site;
A5SS, alternative 5′ splice site; CLF, CURLY LEAF;
COP1, CONSTITUTIVELY
PHOTOMORPHOGENIC1; CUL4, Cullin 4; DAG, day
after germination; DDB1, DNA Damage Binding
Protein 1; E(z), enhancer of zeste; EMB1579,
EMBRYO DEFECTIVE 1579; EMF2, EMBRYONIC
FLOWER 2; ESC, extra sex combs; FIE,
FERTILIZATION INDEPENDENT ENDOSPERM;
FIS2, FERTILIZATION INDEPENDENT SEED; FLC,
*FLOWERING LOCUS C*; FRAP, fluorescence
recovery after photobleaching; H3K27me3,
trimethylation of lysine 27 of histone H3; hnRNP,
heterogeneous nuclear ribonucleoprotein; IDR,
intrinsically disordered protein region; LLPS, liquid-
liquid phase separation; MEA/FIS1, MEDEA; MSI1–
5, MULTIPLE SUPPRESSOR OF IRA 1–5; MXE,
mutually exclusive exon; NLS, nuclear localization
signal; NPM1, nucleophosmin; p55, nucleosome
remodeling factor 55-kDa; PcG, Polycomb group;
PRC1, Polycomb Repressive Complex 1; qRT-PCR,
quantitative reverse transcription PCR; RBP, RNA
binding protein; RFP, red fluorescent protein; RNA-
seq, RNA sequencing; RS, serine/arginine; SIM,
structured illumination microscopy; snRNP, small
nuclear ribonucleoprotein; SPA, SUPPRESSOR OF
PHYA; Su(z)12, suppressor of zeste 12; SWN,
SWINGER; TGFP, tandem copies of enhanced
green fluorescent protein; U snRNP, uridine-rich
snRNP; VRN2, VERNALIZATION 2; WT, wild type.

surrounding environment [1–3]. Among various cellular events, gene transcription and mRNA
splicing play essential roles during plant growth and development as well as during the interac-
tion of the plant with its surrounding environment [4–6]. Dysfunction in gene transcription
and mRNA splicing causes dramatic defects in development and environmental adaptation in
plants [7–11]. Therefore, it is important to understand how plants tightly and efficiently control
gene transcription and mRNA splicing in response to various internal and external cues.

The nucleus contains a dynamic mix of nonmembranous subcompartments, including the
nucleolus, nuclear speckles, paraspeckles, Cajal bodies, nuclear stress bodies, histone locus
bodies, and the perinuclear compartment [12–14]. The absence of a surrounding membrane
enables those subcompartments to assemble or disassemble rapidly following alterations in the
cell's environment and in response to intracellular signals [15–20]. Subnuclear compartmen-
talization might be especially important in mediating rapid changes in gene transcription and
mRNA splicing in response to intrinsic and environmental variations, as proteins associated
with transcription and mRNA splicing are often localized to nuclear speckles or dots [21,22].
Those proteins often exhibit multivalent features that are contributed by repetitive folded
domains and/or disordered regions (also referred to as intrinsically disordered protein regions
[IDRs]) [23], and they are able to undergo liquid-liquid phase separation (LLPS). Indeed, such
proteins have been implicated in the regulation of gene transcription and mRNA splicing in
different organisms [24–29]. However, it remains largely unknown how and to what extent
LLPS of these proteins is linked to gene transcription and mRNA splicing.

Polycomb group (PcG) proteins have been implicated in the regulation of transcription to
establish and maintain specific gene expression patterns to drive organismal development.
PcG proteins form multisubunit protein complexes, such as Polycomb Repressive Complex 1
(PRC1) and PRC2. PRC2 is recruited to target genes and catalyzes the trimethylation of lysine
27 of histone H3 (H3K27me3) [30]. There are four core components of the PRC2 complex,
first identified in *Drosophila*: enhancer of zeste (E[z]); extra sex combs (ESC); suppressor of
zeste 12 (Su[z]12); and nucleosome remodeling factor 55-kDa (p55) [31]. Homologues of
these four core subunits exist in plants. Specifically, *Arabidopsis* has three E(z) homologues,
CURLY LEAF (CLF), MEDEA (MEA/FIS1), and SWINGER (SWN), which catalyze
H3K27me3. In addition, *Arabidopsis* has three Su(z)12 homologues, EMBRYONIC FLOWER
2 (EMF2), VERNALIZATION 2 (VRN2), and FERTILIZATION INDEPENDENT SEED
(FIS2); one Esc homologue, FERTILIZATION INDEPENDENT ENDOSPERM (FIE); and
five p55 homologues, MULTIPLE SUPPRESSOR OF IRA 1–5 (MSI1–5). Based on their differ-
ent subunit compositions, at least three different PRC2-like complexes with distinct functions
exist in *Arabidopsis*: the EMF, VRN, and FIS complexes [32]. Among them, the vegetative
EMF complex, which comprises EMF2, FIE, CLF, or SWN and one p55 homologue, has been
implicated in the regulation of vegetative development and the transition to flowering in *Ara-
bidopsis* [33,34]. Biochemical purification of the EMF complex showed that MSI1, but not
MSI4, was a core subunit [35]. Nevertheless, MSI4 plays a key role in the regulation of floral
transition, which has been linked to the function of the CLF-containing EMF complex
(CLF-PRC2) in repressing the expression of *FLOWERING LOCUS C* (*FLC*) [36]. MSI4 has
also been suggested to play a role in histone deacetylation [37]. Molecular characterization
showed that MSI4 is linked to the epigenetic regulation of the *FLC* locus through its interaction
with Cullin 4 (CUL4)–DNA Damage Binding Protein 1 (DDB1) and a CLF-PRC2 complex
[36]. Specifically, MSI4 is a WD40 repeat-containing protein with a conserved WDxR motif,
which is a typical feature of the previously identified WD40-containing DDB1 and CUL4-asso-
ciated factors [38]. MSI4 forms a complex with DDB1 and CUL4, named CUL4-DDB1$^{MSI4}$
[36]. Although CUL4-DDB1 acts in the photoperiod flowering pathway by interacting with
the CONSTITUTIVELY PHOTOMORPHOGENIC1 (COP1)–SUPPRESSOR OF PHYA

(SPA) complex [39] to control the abundance of CONSTANS protein [40,41], both CUL4 and MSI4 are required to maintain the level of H3K27me3 on *FLC* chromatin [36]. It was also demonstrated that CUL4–DDB1$^{MSI4}$ physically associates with a CLF-PRC2 complex [36]. Therefore, the emerging scenario is that MSI4 forms the CUL4-DDB1$^{MSI4}$ complex, which physically interacts with CLF-PRC2 to establish and/or maintain the level of H3K27me3 on the *FLC* locus to control the expression of *FLC* [36].

Here we report that the plant-specific protein EMBRYO DEFECTIVE 1579 (EMB1579), which was uncovered during the systematic identification of genes required for normal embryo development in *Arabidopsis* [42], is able to undergo LLPS in vitro and in vivo. EMB1579 condensates exhibit liquid-like properties and turn over extremely rapidly within the nucleus. We found that many nuclear proteins crucial for chromosomal function and RNA biology interact with EMB1579 condensates and some of them colocalize with EMB1579 condensates in the nucleus. Loss of function of *EMB1579* alters global gene transcription and mRNA splicing, which provides an explanation for why *emb1579* mutants exhibit pleiotropic developmental defects. Using floral transition as the representative physiological process, we demonstrate that EMB1579 is involved in regulating the level of H3K27me3 on *FLC* and the expression of *FLC*. EMB1579 likely controls the function of CLF-PRC2 via direct interaction with CUL4-DDB1$^{MSI4}$. We propose that EMB1579 condensates condense important biomolecules in the *Arabidopsis* nucleus to regulate their functions in controlling key nuclear events, such as gene transcription and mRNA splicing. Our study thus reveals a new mechanism for the regulation of plant growth and development through LLPS of EMB1579.

## Results

### Loss of function of *EMB1579* induces pleiotropic growth and developmental defects in *Arabidopsis*

*EMB1579* initially caught our attention because it encodes a protein that is homologous to the tobacco protein MAP190, which interacts with both actin filaments and microtubules [43]. It was also proposed to be involved in nuclear calcium signaling during the salt response, as it contains an EF-hand motif [44]. To gain insights into the developmental functions of *EMB1579*, we examined its tissue expression pattern and found that it is widely expressed, especially in highly proliferative tissues, including embryos, the root meristematic region, the basal region of shoots, and the base of the cauline-leaf branch (S1 Fig). To examine the function of *EMB1579*, we characterized two T-DNA insertion lines that were shown to be knockout alleles (Fig 1A). We observed embryonic developmental defects at different stages, resulting from distorted cell division and cell expansion in *emb1579* embryos compared to wild type (WT) (Fig 1B). Specifically, we found that the division plane of cells was mispositioned and cells were swollen in *emb1579* mutants (Fig 1B). Consequently, the development of seeds and seedlings was defective in *emb1579* mutants (Fig 1C–1E). Consistent with the distorted cell phenotypes in embryos, we found that the cell files were altered in the roots of *emb1579* mutants (Fig 1F). Notably, we found that the number of meristematic cells was decreased significantly in *emb1579* mutants compared to WT (Fig 1G). We also found that the floral transition was delayed in *emb1579* mutant plants (Fig 1H and 1I). Thus, our study suggests that *EMB1579* is crucial for *Arabidopsis* growth and development.

### EMB1579 forms highly dynamic liquid-like condensates in the nucleus, and phase separates in vitro

We next generated a functional EMB1579–tandem copies of enhanced green fluorescent protein (TGFP) fusion protein (S2 Fig). We found that it localizes to the nucleus (Fig 2A and 2B)

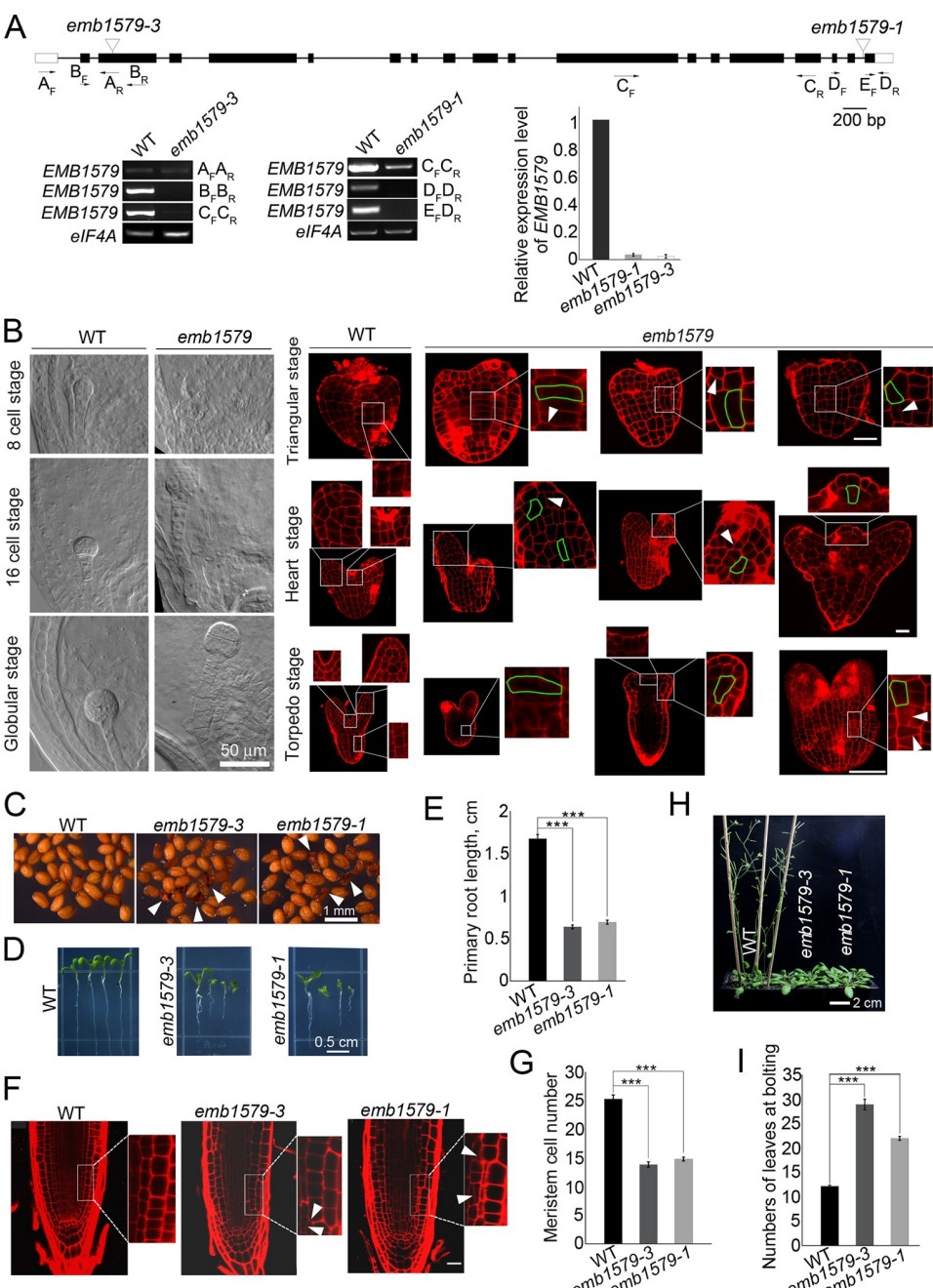

**Fig 1. Loss of function of *EMB1579* induces pleiotropic growth and developmental defects in *Arabidopsis*.** (**A**) Structure of the *EMB1579* gene and identification of T-DNA insertion mutants of *EMB1579*. Two T-DNA insertion lines, CS16026 and Salk_007142, were designated as *emb1579-1* and *emb1579-3*, respectively. The positions of the T-DNA insertions are indicated by inverted triangles. Three independent pairs of primers were used to identify truncated *EMB1579* transcripts in *emb1579-1* and *emb1579-3*. The positions of the primers are indicated under the gene. The expression of *EMB1579* in WT and *emb1579* mutants was also confirmed by qRT-PCR analysis with primer pairs EMB1579-qRT-F1/EMB1579-qRT-R2 (S4 Table). Data are presented as mean ± s.e.m, *n* = 3. Numerical data underlying the graph are available in S1 Data. The original pictures are available in S1 Raw Images. (**B**) Micrographs of embryos at different stages. Embryos at the 8-cell stage, 16-cell stage, and globular stage were revealed by whole-mount clearing methods, and embryos at the triangular stage, heart stage, and torpedo stage were revealed by staining with PI as described previously [45]. In *emb1579* mutants, the swollen cells are outlined with green lines, and white arrowheads indicate the formation of abnormal cell plates. Bars = 50 μm. (**C**) Images of *Arabidopsis* seeds. White arrowheads indicate dry wrinkled seeds. Bar = 1 mm. (**D**) Images of *Arabidopsis* seedlings. Bar = 0.5 cm. (**E**) Quantification of primary root length of 7-day-old seedlings in WT, *emb1579-1*, and *emb1579-3*. Data are presented as

mean ± s.e.m. ***$P$ < 0.001 by Student $t$ test. Numerical data underlying this panel are available in S1 Data. (**F**) Images of *Arabidopsis* roots revealed by staining with PI. White arrowheads indicate formation of abnormal cell plates. Bar = 25 μm. (**G**) Quantification of meristem cell number of 3-day-old seedling roots in WT, *emb1579-1*, and *emb1579-3*. Data are presented as mean ± s.e.m. ***$P$ < 0.001 by Student $t$ test. Numerical data underlying this panel are available in S1 Data. (**H**) Images of 6-week-old *Arabidopsis* plants. Bar = 2 cm. (**I**) Quantification of the number of rosette leaves at bolting in WT, *emb1579-1*, and *emb1579-3*. Data are presented as mean ± s.e.m. ***$P$ < 0.001 by Student $t$ test. Numerical data underlying this panel are available in S1 Data. EMB1579, EMBRYO DEFECTIVE 1579; PI, propidium iodide; qRT-PCR, quantitative reverse transcription PCR; WT, wild type.

and undergoes dynamic changes during the cell cycle (Fig 2C, S1 Movie). Specifically, the intense nuclear signal of EMB1579-TGFP disappeared with the breakdown of the nuclear membrane. The nuclear localization of EMB1579 is consistent with the presence of a nuclear localization signal (NLS) in the protein sequence [46]. We found that EMB1579 forms compartments within the nucleus of *Arabidopsis* root cells (Fig 2D) and the size of the compartments differs between proliferative and differentiated *Arabidopsis* root cells (Fig 2D and 2E). EMB1579 compartments are distinct from Cajal bodies and HYL1 bodies (S3 Fig). Strikingly, we found that EMB1579 compartments are extremely dynamic, as assayed with fluorescence recovery after photobleaching (FRAP) experiments (Fig 2F and 2G, S3 Movie), and EMB1579 compartments are able to undergo fusion (Fig 2H, S2 Movie). These data suggest that EMB1579 compartments exhibit liquid-like properties in the nucleus.

Next, we wondered whether EMB1579 on its own can phase separate in vitro. We initially analyzed the amino acid sequence of EMB1579 using different phase-separation predictors reported previously [47,48] and found that it contains IDRs that occupy most of the protein (Fig 2I). In addition, we found that EMB1579 contains few hydrophobic residues (Fig 2I), which may lead to weak hydrophobic interactions that will facilitate LLPS to drive the formation of compartments as shown previously [49,50]. Indeed, we found that full-length recombinant EMB1579 protein (Fig 2J) forms highly spherical liquid-like condensates (Fig 2K). Strikingly, the condensates are able to fuse together (Fig 2K, S4 Movie). In particular, aqueous solutions of EMB1579 spontaneously form condensates at high protein concentrations in a dose-dependent manner, even when the salt concentration is comparatively high (Fig 2L). EMB1579 is able to undergo LLPS at 25 nM (Fig 2L), which suggests that the critical concentration for EMB1579 to undergo LLPS is very low. Interestingly, we estimated that the concentration of EMB1579 in the nucleus is about 33 nM (S4 Fig), which suggests that EMB1579 on its own can undergo LLPS in the nucleus. FRAP experiments showed that EMB1579 within condensates dynamically exchanged with EMB1579 in the surrounding environment (Fig 2M), with an average half-time of recovery of 2.61 seconds (Fig 2N). We next performed the experiment described as "half-bleach" [18,51,52] to test whether the components within a liquid condensate undergo constant mixing. We bleached roughly half of an EMB1579 condensate and monitored its fluorescence over time. We found that EMB1579 rapidly redistributed from the unbleached area to the bleached area (Fig 2O, S5 Movie), which suggests that EMB1579 molecules can diffuse freely within condensates and they are in a liquid state. These data together suggest that EMB1579 undergoes LLPS in vitro.

## Many nuclear proteins bind to EMB1579 condensates, and loss of function of *EMB1579* affects global transcription and mRNA splicing in *Arabidopsis*

In order to understand how EMB1579 performs its cellular functions, we next asked which proteins enter EMB1579 condensates. Using the conditions under which EMB1579 phase separates in vitro, we incubated EMB1579 with an *Arabidopsis* root total extract and performed pull-down experiments followed by mass spectrometry. We found that many nuclear proteins

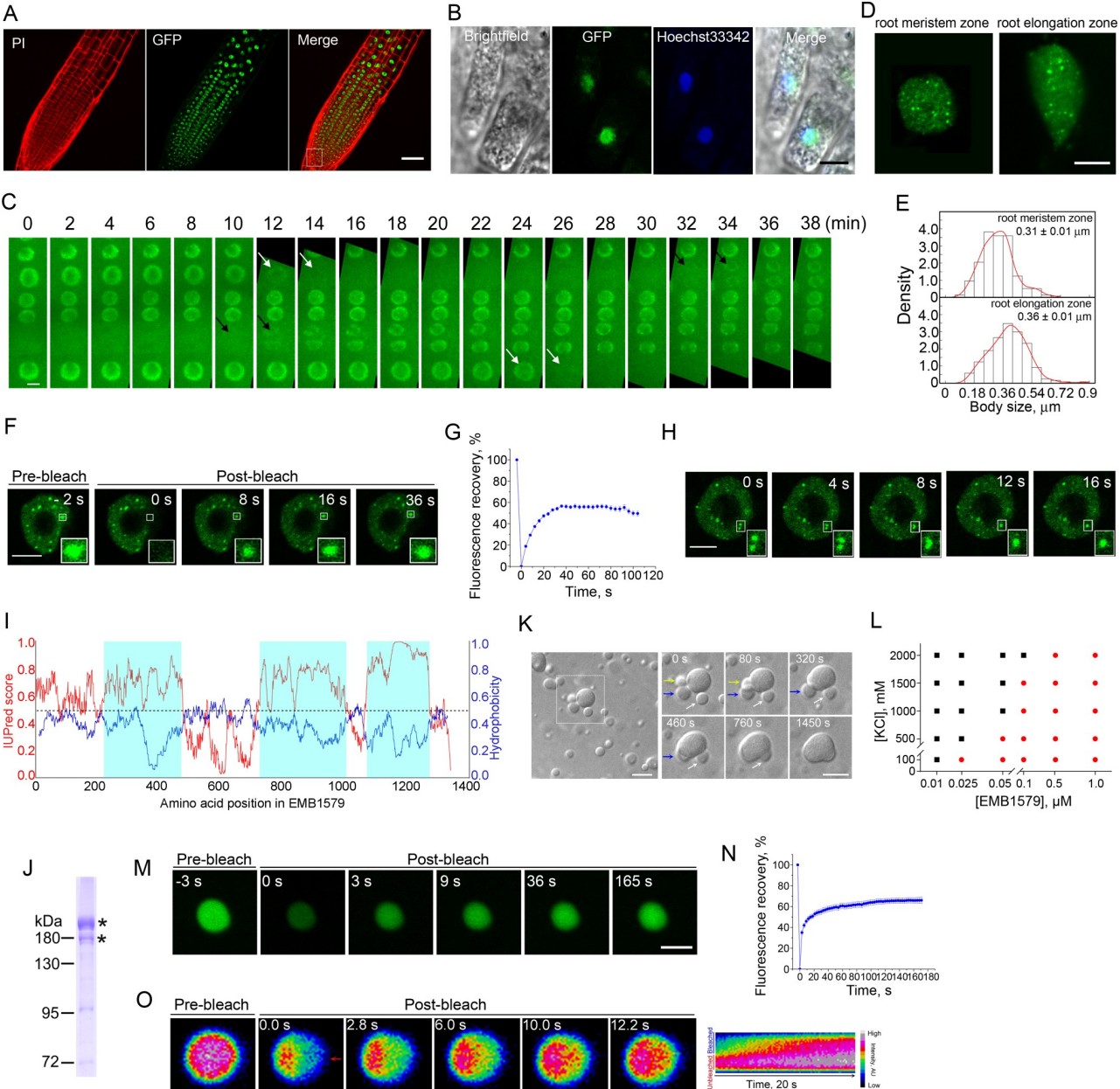

**Fig 2. EMB1579 forms dynamic bodies in the nucleus of *Arabidopsis* root cells and undergoes rapid phase separation in vitro.** (**A**) Micrographs of *Arabidopsis* primary root from pCAMBIA1301-proEMB1579::gEMB1579-TGFP; *emb1579* plants. Root cells were revealed by staining with PI. Bar = 50 μm. (**B**) Micrographs of *Arabidopsis* root cells showing the subcellular localization of EMB1579. Nuclei were stained with Hoechst 33342. Bar = 5 μm. (**C**) Time-lapse images of root cells from pCAMBIA1301-proEMB1579::gEMB1579-TGFP; *emb1579*. White arrows indicate the time points that EMB1579 starts to disappear from the nucleus, whereas black arrows indicate the time points that EMB1579 starts to appear in the nucleus. Bar = 5 μm. (**D**) EMB1579 forms bodies within the nucleus of *Arabidopsis* root cells from the meristem zone and the elongation zone. Bar = 5 μm. (**E**) Histograms of size distribution of EMB1579 bodies in the nucleus from cells in the root meristem zone and the elongation zone. More than 200 bodies were measured in at least 12 nuclei. The average values are presented as mean ± s.e.m. Numerical data underlying this panel are available in S1 Data. (**F**) Images of an *Arabidopsis* nucleus before and after bleaching. The photobleached region is boxed. Bar = 5 μm. (**G**) Plot of fluorescence intensity before and after photobleaching. The blue curve represents the average value of fluorescence intensity of 18 bodies from 10 seedlings. All data are presented as mean ± s.e.m. Numerical data underlying this panel are available in S1 Data. (**H**) Images of an *Arabidopsis* nucleus showing the fusion of two EMB1579 bodies. Boxed region indicates two bodies that undergo fusion. Bar = 5 μm. (**I**) Analysis of the intrinsic disorder tendency and hydropathicity of the full-length EMB1579 protein. The intrinsic disorder (red line) was analyzed with IUPred2A. The scores are assigned between 0 and 1, and a score above 0.5 indicates disorder. Three long stretches of disordered regions are shaded in light blue. The hydropathicity score (blue line) was determined with ExPASy, which used the Kyte-Doolittle scale of amino acid hydropathicity with a sliding window size of 21. The scores were normalized from 0 to 1. A score above 0.5 indicates hydrophobicity. Numerical data underlying this panel are available in S1 Data. (**J**) Purified

recombinant EMB1579 protein. The asterisks indicate EMB1579 protein bands. The lower band might be a degradation product of the full-length EMB1579. The original pictures are available in S1 Raw Images. (**K**) EMB1579 condensates visualized by DIC optics. The right panels show time-lapse images of the boxed region in the left panel. Condensates fuse to form a single large condensate. The arrows indicate the fusion events. Bars = 10 μm. (**L**) Phase diagram of EMB1579 condensate formation at the indicated protein and salt concentrations. The diagram was plotted after scoring the optically resolvable droplets at different protein/KCl concentrations. Red circles indicate the formation of condensates; black squares indicate no formation of condensates. (**M**) Images of EMB1579 droplets labeled with Oregon Green before and after photobleaching. Bar = 2 μm. (**N**) Plot of the changes in fluorescence intensity during the FRAP experiment. The blue curve represents the average value of fluorescence intensity from 15 bodies. Values are presented as mean ± s.e.m. Numerical data underlying this panel are available in S1 Data. (**O**) Half-FRAP of EMB1579 condensates. The left panel shows images of EMB1579 condensates labeled with Oregon Green before and after photobleaching during the half-bleach experiment. The right panel shows the kymograph analysis for the unbleached and bleached regions. Bar = 1 μm. EMB1579, EMBRYO DEFECTIVE 1579; DIC, differential interference contrast; FRAP, fluorescence recovery after photobleaching; TGFP, tandem copies of enhanced green fluorescent protein; PI, propidium iodide; WT, wild type.

crucial for chromosomal function and RNA biology are enriched in the pull-down fraction (Fig 3A, S1 Table), which suggests that EMB1579 might function by regulating the activities of those proteins. Next, we determined whether the transcriptional profile of the genes encoding those proteins was altered in *emb1579* mutants. We performed RNA sequencing (RNA-seq) analysis on 7-day-old *Arabidopsis* seedlings and found that loss of function of *EMB1579* caused differential expression of many genes involved in essential processes, including chromosome organization, DNA and RNA binding, transcription, histone binding and modification, cyto-skeletal organization, and so on (Fig 3B, S2 Table). The differential expression of 17 previously characterized genes was validated by quantitative reverse transcription PCR (qRT-PCR) analysis (Fig 3C). To determine whether pre-mRNA splicing events were altered in *emb1579* mutants, we searched our RNA-seq data for splicing defects and found that there was a total of 2,507 abnormal splicing events in *emb1579* mutants, which could be divided into five different patterns (Fig 3D, S3 Table). Among them, skipped exons and retained introns were most common in *emb1579* mutants (Fig 3D). We validated the splicing defects in three previously characterized genes. We showed that transcripts of the flowering repressor gene *FLC* and the cell division inhibition gene *ICK2* [53] have intron splicing defects (Fig 3E–3I), and transcripts of the cell cycle promotion gene *CYCD2;1* [54] have a skipped exon in *emb1579* (Fig 3J–3L). The defects in global gene transcription and pre-mRNA splicing provide an explanation for why *emb1579* mutants exhibit pleiotropic growth and developmental defects. In support of the notion that the gene transcription and pre-mRNA splicing defects account for the developmental defects in *emb1579* mutants, we found that down-regulation of *FLC*, which has both transcriptional and mRNA splicing defects in *emb1579* mutants (Fig 3C, 3E and 3F), can suppress the later flowering phenotype in *emb1579* mutants (S5 Fig). In summary, we demonstrate that EMB1579 is involved in the regulation of transcription and pre-mRNA splicing in *Arabidopsis*.

## Some proteins crucial for gene transcription and mRNA splicing colocalize with EMB1579 condensates in the nucleus

We next determined whether specific proteins crucial for gene transcription and mRNA splicing can enter EMB1579 condensates in vivo. We found that MSI4 and DDB1A appeared in the EMB1579 condensates pull-down fraction (Fig 3A, S1 Table). MSI4/FVE is known to be a key regulator of the autonomous flowering pathway that constitutively reduces the expression of *FLC* [36]. MSI4 interacts with CUL4-DDB1 and a PRC2-like complex to mediate this function [36]. Therefore, we performed colocalization experiments to determine whether MSI4 and DDB1B (homologue of DDB1A) can enter EMB1579 condensates. Both proteins do indeed colocalize with EMB1579 condensates in vivo (Fig 4A). Spliceosome components, including uridine-rich small nuclear ribonucleoproteins (U snRNPs), non-snRNP factors, serine/

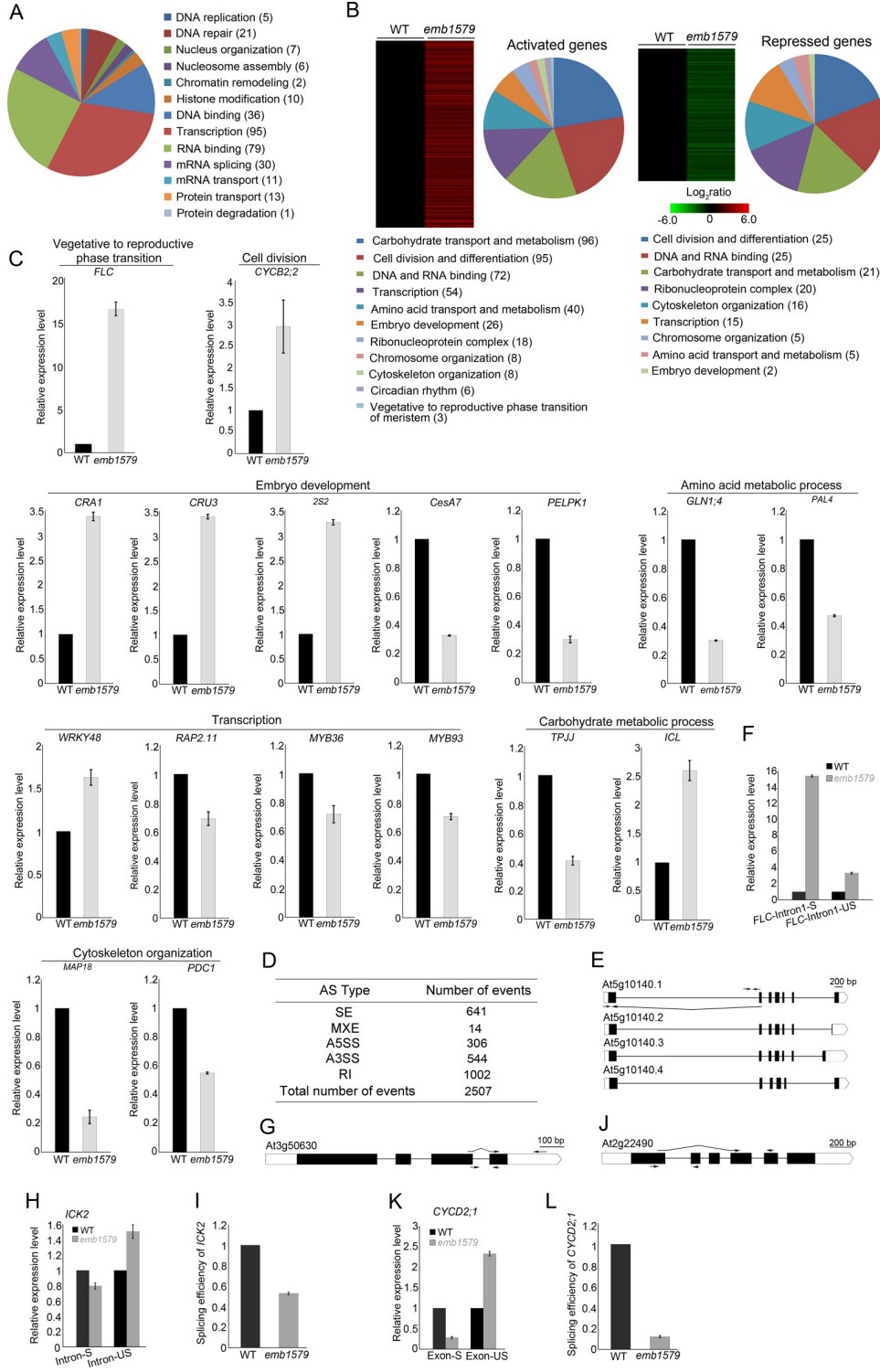

**Fig 3. Loss of function of *EMB1579* alters global transcription and mRNA splicing.** (**A**) Identification and functional classification of proteins that cosediment with EMB1579 compartments. Proteins enriched in the EMB1579 cosedimentation fraction were analyzed by mass spectrometry. A full list of the proteins is presented in S1 Table. (**B**) EMB1579-activated and EMB1579-repressed genes. Seven-DAG WT and *emb1579* seedlings were subjected to RNA-seq analysis. A full list of the up- and down-regulated genes is presented in S2 Table. (**C**) Validation of the altered expression of specific genes in *emb1579* mutants. qRT-PCR was performed to confirm the changed expression levels of 17 different genes in *emb1579* mutants, as originally revealed by RNA-seq analysis. Data are presented as mean ± s.e.m,

*n* = 3. The selected genes were demonstrated previously to be involved in different physiological processes, as indicated in (**B**). The underlying numerical data are available in S1 Data. (**D**) Summary of splicing defects events in *emb1579* mutant plants. A full list of the splicing defects is presented in S3 Table. (**E**) Schematic representation of four spliced transcripts of *FLC* in WT plants. The white boxes indicate the 5′ UTR and 3′ UTR of *FLC*. The black boxes represent exons and the black lines indicate introns. The positions of primers are indicated by arrows. (**F**) qRT-PCR analysis of the mature *FLC* transcript with the first intron spliced (*FLC*-intron1-S) using the lower primer pair in (**E**) and the immature *FLC* transcript with the first intron retained (*FLC*-intron1-US) using the upper primer pair in (**E**) in WT and *emb1579*. Data are presented as mean ± s.e.m, *n* = 3. Numerical data underlying this panel are available in S1 Data. (**G**) Schematic representation of the *ICK2* (At3g50630) gene structure. The positions of primers are indicated by arrows. (**H**) qRT-PCR analysis of the mature *ICK2* transcript (Intron-S) using the upper primer pair in (**G**) and the immature *ICK2* transcript with an RI (Intron-US) using the lower primer pair in (**G**) in WT and *emb1579*. Data are presented as mean ± s.e.m, *n* = 3. The underlying numerical data are available in S1 Data. (**I**) Examination of the splicing efficiency of the third intron of *ICK2* in *emb1579* mutants. Data are presented as mean ± s.e.m, *n* = 3. The underlying numerical data are available in S1 Data. (**J**) Schematic representation of the *CYCD2;1* (At2g22490) gene structure. The positions of primers are indicated by arrows. (**K**) qRT-PCR analysis of an abnormal *CYCD2;1* transcript (Exon-S) with the upper primer pair in (**J**) and the normal *CYCD2;1* transcript (Exon-US) with the lower primer pair in (**J**) in WT and *emb1579*. Data are presented as mean ± s.e.m, *n* = 3. The underlying numerical data are available in S1 Data. (**L**) Examination of the splicing efficiency of the second and third exons of *CYCD2;1* in *emb1579* mutants. Data are presented as mean ± s.e.m, *n* = 3. The underlying numerical data are available in S1 Data. A3SS, alternative 3′ splice site; A5SS, alternative 5′ splice site; *emb1579*, embryo defective 1579; DAG, day after germination; *FLC*, *FLOWERING LOCUS C*; MXE, mutually exclusive exon; qRT-PCR, quantitative reverse transcription PCR; RI, retained intron; RNA-seq, RNA sequencing; SE, skipped exon; WT, wild type.

arginine (RS) proteins, and heterogeneous nuclear ribonucleoproteins (hnRNPs), are known to play important roles in mRNA splicing [5,55,56]. We selected U1-70K and SC35 as representative U1 snRNP and non-snRNP factors [57–59], respectively, and found that they colocalize with EMB1579 condensates (Fig 4A). RZ-1C, the hnRNP-like RNA binding protein (RBP) in plants, was reported to be involved in the regulation of mRNA splicing [5,11]. We examined the spatial association of RZ-1C with EMB1579 and found that they colocalize in cells (Fig 4A). Thus, we demonstrate that interactors of PRC2 and some spliceosome factors colocalize with EMB1579 condensates in vivo. To determine whether the colocalized or pulled-down proteins physically interact with EMB1579, we performed the firefly split luciferase complementation imaging assay and found that 18 of them do indeed interact with EMB1579 (Fig 4B). Taking these data together, we propose that EMB1579 undergoes LLPS to form liquid-like condensates that condense biomolecules crucial for chromosomal function and RNA biology to control nuclear events, such as transcription and pre-mRNA splicing (Fig 4C).

## The RED repeat is required for the formation of normal-sized EMB1579 condensates and the cellular functions of EMB1579

To directly link the formation of EMB1579 condensates with the functions of EMB1579 in vivo, we wanted to identify the motifs within EMB1579 that facilitate its LLPS and formation of liquid-like condensates. The RED repeat in EMB1579 (Fig 5A) caught our attention because some proteins containing RE, RD, or RED repeats were previously shown to form bodies in cells [60–62]. To determine whether the RED repeat contributes to the formation of EMB1579 condensates, we deleted the RED repeat from the EMB1579 protein to create EMB1579ΔRED (Fig 5B). At all the concentrations tested, EMB1579ΔRED condensates are significantly smaller than EMB1579 condensates in vitro (Fig 5C and 5D), which suggests that the RED repeat is required for EMB1579 to form normal-sized condensates. Furthermore, a higher concentration of EMB1579ΔRED is required to form liquid condensates compared to EMB1579 at the same salt concentration (Fig 5E), which suggests that the RED repeat might modulate the interaction between EMB1579 molecules. Interestingly, we found that EMB1579ΔRED condensates are even more dynamic than EMB1579 condensates (Fig 5F and 5G). Although we do not currently know the reason for this, it could be that the RED repeat, which is enriched in

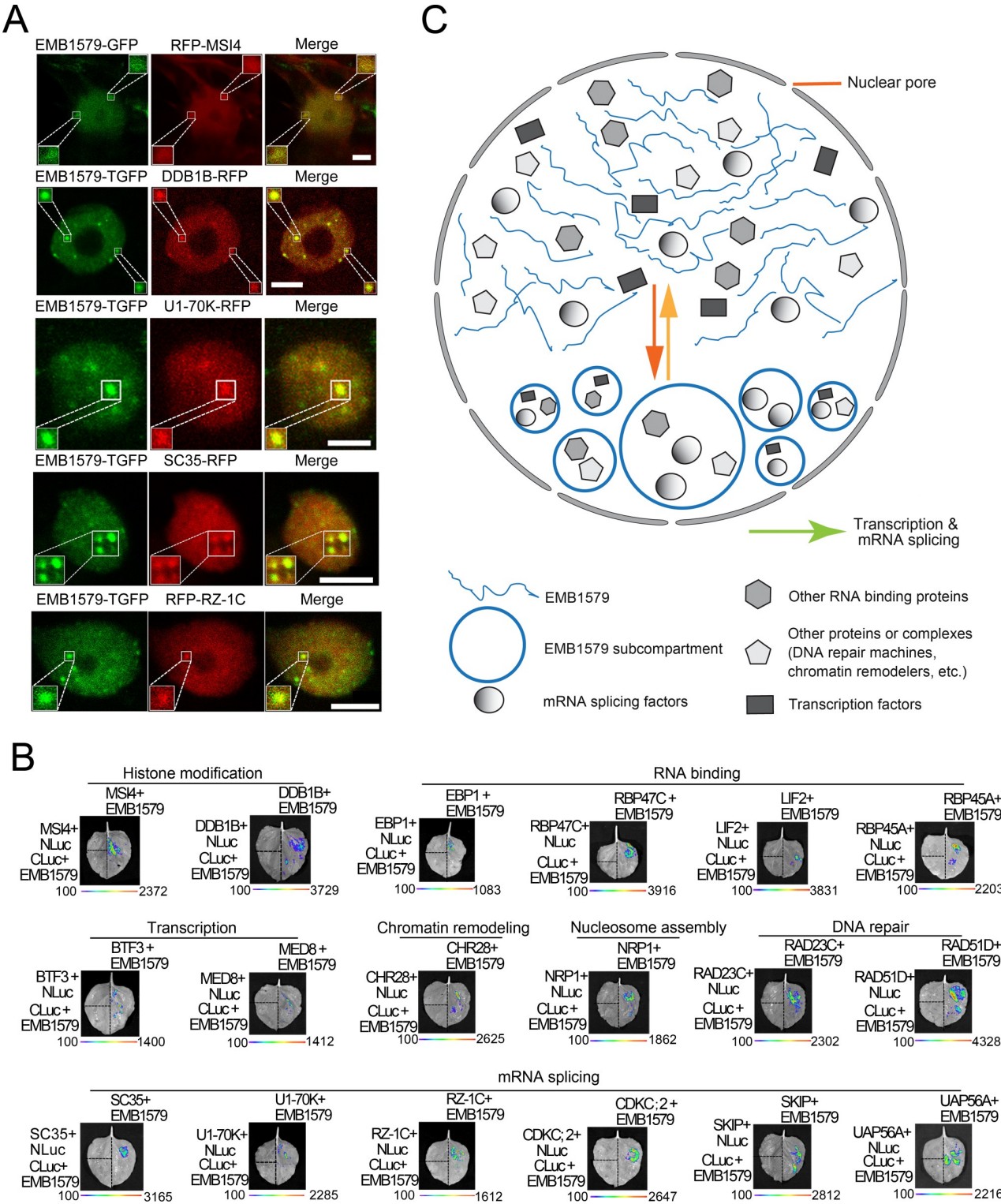

**Fig 4. EMB1579 interacts with specific nuclear proteins, and EMB1579 condensates colocalize with some of them in the nucleus.** (**A**) Images showing colocalization of some nuclear proteins with EMB1579 condensates. The colocalization experiment of MSI4 with EMB1579 was performed in leaf epidermal cells of *Nicotiana benthamiana*, and the colocalization experiments of other nuclear protein with EMB1579 were performed in *Arabidopsis* root cells. Bars = 5 μm. (**B**) Firefly split luciferase complementation imaging assay confirms that 15 nuclear proteins identified by mass spectrometry analysis (Fig 3A) and three colocalized proteins directly interact with EMB1579. (**C**) Simple model for the phase separation of EMB1579 in the nucleus and its potential functions in *Arabidopsis*. EMB1579 undergoes phase separation to form dynamic compartments in the nucleus. The

compartments recruit and concentrate different proteins and/or protein complexes crucial for DNA and RNA biology and consequently control important biochemical reactions, such as transcription and mRNA splicing. DDB1, DNA Damage Binding Protein 1; EMB1579, EMBRYO DEFECTIVE 1579; GFP, green fluorescent protein; MSI4, MULTIPLE SUPPRESSOR OF IRA 4; RFP, red fluorescent protein; TGFP,tandem copies of enhanced GFP.

charged residues, strengthens the electrostatic interaction between EMB1579 molecules (Fig 5A). Nonetheless, these data together suggest that the RED repeat modulates the formation of normal-sized EMB1579 condensates in vitro. We next introduced *EMB1579ΔRED-TGFP* into *emb1579* mutants with its expression under the control of the native *EMB1579* promoter and selected the transgenic plants contain similar amount of EMB1579ΔRED-TGFP when compared to EMB1579-TGFP for subsequent analyses (S6A and S6B Fig). We found that EMB1579-TGFP clearly formed condensates whereas EMB1579ΔRED-TGFP hardly formed any large condensates in cells visualized by laser scanning confocal microscopy (Fig 5H and 5I) or superresolution structured illumination microscopy (SIM) (Fig 5J). This suggests that the RED motif is required for EMB1579 to form normal-sized condensates in vivo. Consistent with this, we found that EMB1579ΔRED-TGFP only partially restored the transcript levels of several genes compared to EMB1579-TGFP (Fig 5K). We also found that EMB1579ΔRED-TGFP, in contrast to EMB1579-TGFP, failed to rescue the primary root growth and floral transition phenotypes in *emb1579* mutants (S6C–S6F Fig). However, we found that EMB1579ΔRED retained the capability to interact with the functionally relevant interactors (S7 Fig). These data together suggest that the cellular functions of EMB1579 likely depend on the formation of appropriate liquid-like compartments.

## EMB1579 interacts with MSI4 and *emb1579* phenocopies *msi4* in terms of flowering time and the level of H3K27m3 on *FLC*

Above, we showed that MSI4 colocalizes with EMB1579 condensates in the nucleus (Fig 4A) and EMB1579 physically interacts with MSI4 (Fig 4B). To understand how exactly *EMB1579* performs its cellular functions, we wanted to determine how *EMB1579* coordinates with MSI4 to control the expression of *FLC*. The physical interaction between EMB1579 and MSI4 was further confirmed by yeast two-hybrid assay (Fig 6A). Next, we found that although both *emb1579* and *msi4* mutants exhibit a late-flowering phenotype (Fig 1H and 1I), this phenotype is weaker in *emb1579* than in *msi4* (Fig 6B and 6C). This allows us to speculate that EMB1579 may act as a regulator of MSI4 in controlling the expression of *FLC* and flowering. It was previously proposed that MSI4 acts by interacting with a PRC2-like complex to establish and/or maintain the levels of H3K27me3 at the *FLC* locus [36]. Considering that the global level of H3K27me3 is reduced in *msi4* mutants [36], we determined the global level of H3K27me3 in *emb1579* mutants and found that it is reduced (Fig 6D). To specifically link the function of EMB1579 to the expression of *FLC*, we analyzed the level of H3K27me3 on *FLC* and found that it is reduced in *emb1579* mutants (Fig 6E). Again, this is in agreement with the previous report that *msi4* mutants have reduced levels of H3K27me3 on *FLC* [36]. Thus, we showed that EMB1579 physically interacts with MSI4 and *emb1579* phenocopies *msi4* in terms of flowering time and the level of H3K27me3 on the *FLC* locus.

## EMB1579 condensates recruit and condense MSI4 in vitro and in vivo

Above, we showed that deletion of the RED repeat of EMB1579 impairs the repression of *FLC* transcription (Fig 5K) and affects flowering time (S6E Fig). These data suggest that the formation of normal-sized liquid-like condensates is crucial for this functional role of EMB1579. Considering that MSI4 colocalizes with EMB1579 condensates in the nucleus (Fig 4A), we speculated that EMB1579 might regulate the function of MSI4 by recruiting and condensing it

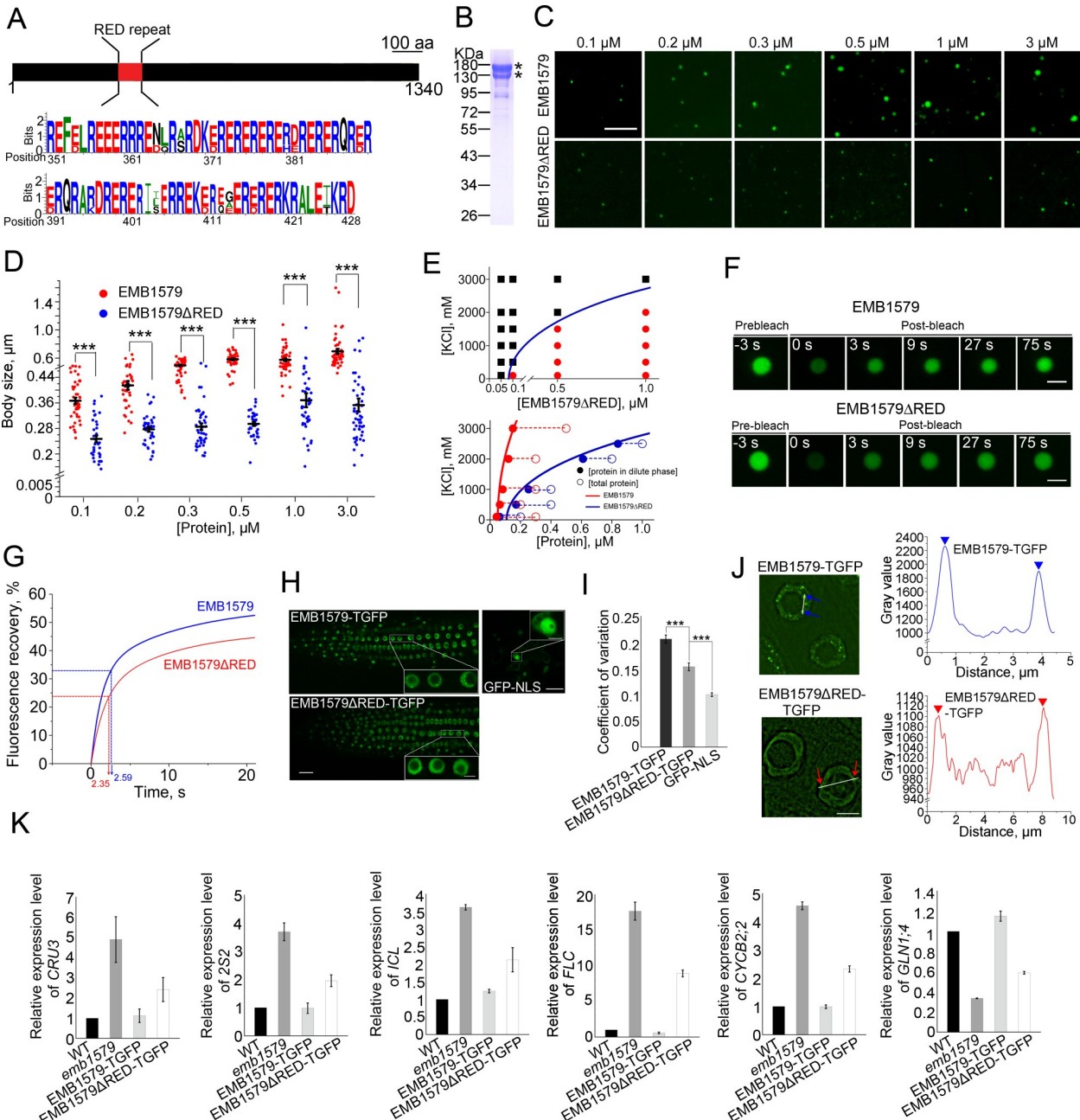

**Fig 5. The RED repeat is crucial for the phase-separation property of EMB1579 in vitro and in vivo.** (**A**) EMB1579 contains a RED repeat. The upper schematic shows the domain organization of EMB1579 and the lower panel is the sequence of the predicted RED motif in EMB1579, analyzed by WEBLOGO (http://weblogo.berkeley.edu/logo.cgi). (**B**) SDS-PAGE analysis of recombinant EMB1579ΔRED. The asterisks indicate EMB1579ΔRED protein bands. The original pictures are available in S1 Raw Images. (**C**) Representative fluorescence images of EMB1579 and EMB1579ΔRED condensates formed under different concentrations. [EMB1579] and [EMB1579ΔRED] range from 0.1 to 3 μM. LLPS reactions were performed in F-buffer containing 100 mM KCl in the absence of PEG 3350. Bar = 5 μm. (**D**) Quantification of the body size of EMB1579 and EMB1579ΔRED condensates. Column scatter chart shows the body size in reactions containing [EMB1579] 0.1 μM (*n* = 42) and [EMB1579ΔRED] 0.1 μM (*n* = 35), [EMB1579] 0.2 μM (*n* = 37) and [EMB1579ΔRED] 0.2 μM (*n* = 39), [EMB1579] 0.3 μM (*n* = 41) and [EMB1579ΔRED] 0.3 μM (*n* = 37), [EMB1579] 0.5 μM (*n* = 39) and [EMB1579ΔRED] 0.5 μM (*n* = 45), [EMB1579] 1 μM (*n* = 51) and [EMB1579ΔRED] 1 μM (*n* = 41), [EMB1579] 3 μM (*n* = 47) and [EMB1579ΔRED] 3 μM (*n* = 50). Data are presented as mean ± s.e.m. ***$P < 0.001$ by Student *t* test. Numerical data underlying this panel are available in S1 Data. (**E**) The capability of EMB1579ΔRED to form condensates is impaired in vitro. The upper panel shows the phase diagram of EMB1579ΔRED condensate formation at the indicated protein and KCl concentrations. The phase diagram was plotted as described in Fig 2L. Red circles indicate the formation of droplets; black squares indicate no formation of condensates. The blue line indicates the phase boundary of EMB1579ΔRED. The lower panel shows the saturation curves for the phase separation of EMB1579 and EMB1579ΔRED. The protein concentration in the dilute phase (solid circles) is plotted for various total protein concentrations (empty circles) at five different KCl

concentrations for EMB1579 and EMB1579ΔRED. The same colored solid and empty circles represent the data from the same set of experiments. The concentration of the dilute phase for EMB1579 and EMB1579ΔRED falls directly onto the phase boundary of EMB1579 and EMB1579ΔRED (solid lines) for all conditions. The underlying numerical data are available in S1 Data. (**F**) FRAP analysis of EMB1579 and EMB1579ΔRED condensates. Bars = 2 μm. (**G**) Plot of the fluorescence intensity during FRAP experiments. The blue curve shows the average fluorescence intensity from 15 EMB1579 bodies and the red curve represents the average fluorescence intensity from 20 EMB1579ΔRED bodies. Data are presented as the mean. The dashed lines represent $t_{1/2}$ of EMB1579 and EMB1579ΔRED. Numerical data underlying this panel are available in S1 Data. (**H**) No obvious compartments are formed by EMB1579ΔRED-TGFP compared to EMB1579-TGFP. GFP-NLS is a control that shows diffuse fluorescence in the nucleoplasm. Bar = 20 μm for the whole image and bar = 5 μm for the inset image. (**I**) Analysis of the coefficient of variation of fluorescence of GFP, EMB1579-TGFP, and EMB1579ΔRED-TGFP. Data are presented as mean ± s.e.m. ***$P < 0.001$ by Student $t$ test. More than 30 nuclei were measured from 20 seedlings. Numerical data underlying this panel are available in S1 Data. (**J**) SIM images of nuclei harboring EMB1579-TGFP or EMB1579ΔRED-TGFP in *Arabidopsis* root cells. The bodies formed by EMB1579-TGFP and EMB1579ΔRED-TGFP are indicated by blue and red arrows, respectively. Bar = 5 μm. The gray values of EMB1579-TGFP (blue) and EMB1579ΔRED-TGFP (red) were measured along the lines shown in the left panel. The triangles indicate the peaks of fluorescence. The underlying numerical data are available in S1 Data. (**K**) qRT-PCR analysis to determine the transcript levels of six genes in WT, *emb1579*, and complementation lines. Data are presented as mean ± s.e.m, $n = 3$. The underlying numerical data are available in S1 Data. aa, amino acid; EMB1579, EMBRYO DEFECTIVE 1579; FRAP, fluorescence recovery after photobleaching; GFP, green fluorescent protein; LLPS, liquid-liquid phase separation; NLS, nuclear localization signal; qRT-PCR, quantitative reverse transcription PCR; SIM, structured illumination microscopy; TGFP, tandem copies of enhanced GFP; WT, wild type.

into the condensates. Indeed, we found that EMB1579 condensates can recruit and condense recombinant MSI4 protein in vitro (Fig 6F and 6G). Next, we incubated EMB1579 with *Arabidopsis* root total extract containing red fluorescent protein (RFP)-MSI4 under the conditions that induce phase separation of EMB1579 in vitro. We found that EMB1579 condensates actively recruit and condense RFP-MSI4 in this system (Fig 6H), which suggests that EMB1579 condensates might be able to condense MSI4 in vivo. In support of this speculation, we found that loss of function of *EMB1579* impaired the capability of RFP-MSI4 to form compartments in *Arabidopsis* protoplasts (Fig 6I). These data together suggest that EMB1579 condensates can recruit and condense MSI4.

## EMB1579 interacts with CUL4 and DDB1 but not CLF and FIE

MSI4 regulates the flowering time by interacting with CUL4-DDB1 [36]. Furthermore, MSI4 and DDB1 appeared in the EMB1579 pull-down fractions (Fig 3A, S1 Table), and they both colocalized with EMB1579 condensates in vivo (Fig 4A). This prompted us to ask how EMB1579 may functionally coordinate with CUL4-DDB1$^{MSI4}$ complex to maintain the level of H3K27me3 on *FLC* and repress the expression of *FLC*. We used the firefly split luciferase complementation imaging assay to show that EMB1579 interacts with CUL4 *in planta* (S8A Fig). This result, together with our finding that EMB1579 interacts with MSI4 and DDB1B *in planta* (Fig 4B), suggests that EMB1579 interacts with MSI4, DDB1, and CUL4. The interactions were further confirmed by the yeast two-hybrid assay. Specifically, we found that MSI4 can interact with full-length EMB1579 (S8B Fig), and a fragment of EMB1579 containing the MSI4-binding domain can bind to CUL4 (S8C and S8D Fig). Interestingly, we found that the RED repeat is not required for the interaction between EMB1579 and MSI4 (S8C Fig). MSI4 regulates the flowering time by forming CUL4-DDB1$^{MSI4}$ complex that interacts with a CLF-PRC2 complex to repress the expression of *FLC* via H3K27 methylation [36], and loss of function of *EMB1579* reduces the level of H3K27me3 on *FLC* chromatin. Therefore, we wondered whether EMB1579 physically interacts with CLF and FIE, core subunits of CLF-PRC2. We found that EMB1579 did not interact with CLF and FIE (S8A, S8B and S8D Fig). These data together suggest that EMB1579 regulates the level of H3K27me3 on *FLC* chromatin via interaction with CUL4-DDB1$^{MSI4}$ complex rather than direct interaction with CLF-PRC2. As EMB1579 condensates can recruit and condense MSI4 both in vitro and in vivo, we wondered whether EMB1579 condensates can also recruit and condense DDB1 and CUL4. We thus performed the in vitro recruitment experiments using DDB1B as the representative protein, since it colocalizes with EMB1579 condensates in vivo (Fig 4A). We found that EMB1579 condensates can

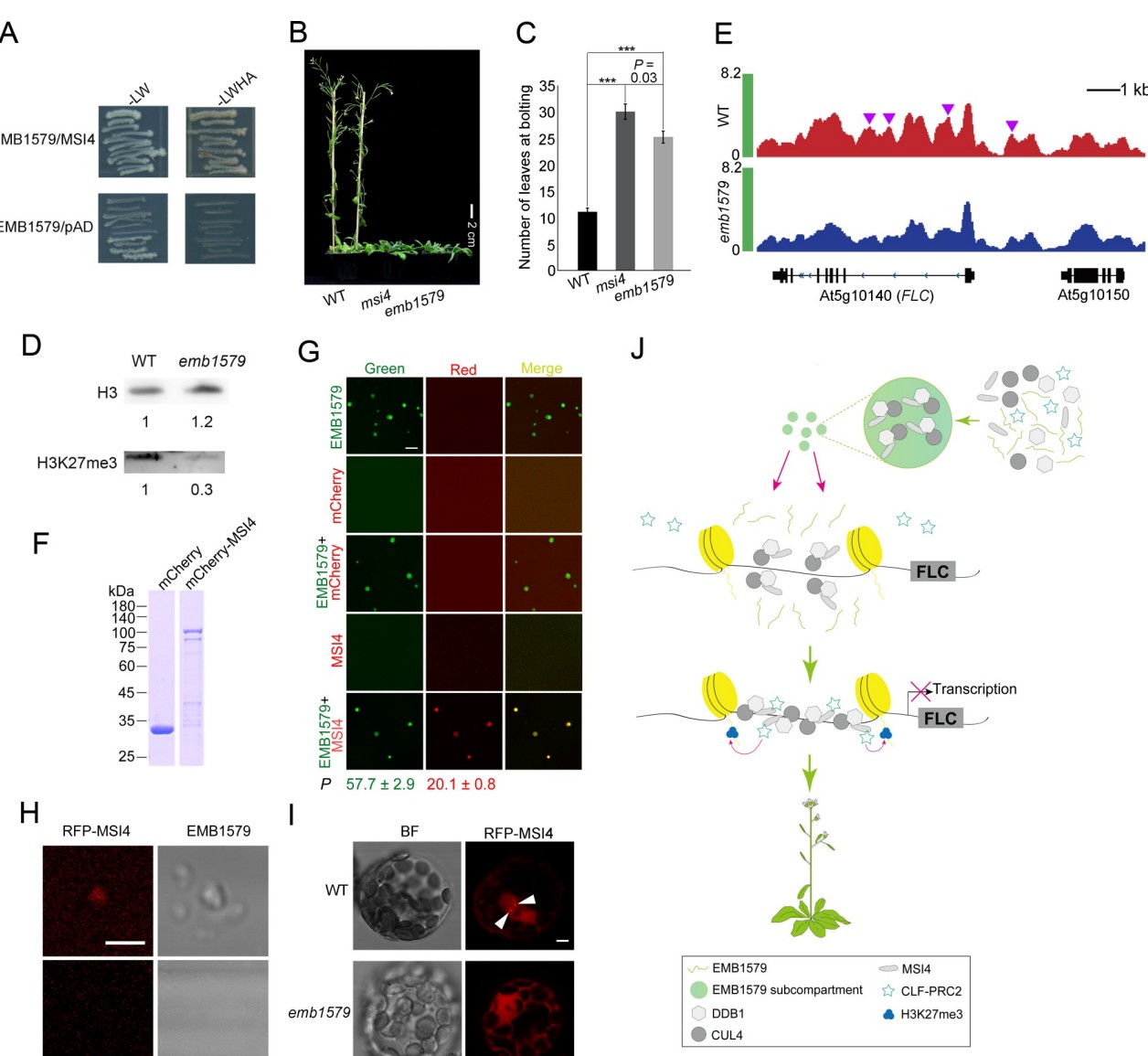

**Fig 6. Phenotypic similarities between *emb1579* and *msi4*, and EMB1579 condensates can condense MSI4 in vitro and in vivo.** (**A**) Yeast two-hybrid analysis of the interaction of MSI4 with EMB1579. (**B**) Images of 7-week-old *Arabidopsis* plants. WT, *msi4* mutants, and *emb1579* mutants are shown. Bar = 2 cm. (**C**) Quantification of the number of rosette leaves at bolting in WT, *msi4*, and *emb1579*. Data are presented as mean ± s.e.m. ***$P < 0.001$ by Student $t$ test. Numerical data underlying this panel are available in S1 Data. (**D**) Analysis of the global level of H3K27me3 in WT and *emb1579* seedlings. The western blot of total nuclear proteins was probed with anti-H3 (top panel) and anti-H3K27me3 (bottom panel) antibodies. The original pictures are available in S1 Raw Images. (**E**) Representative genome browser view of H3K27me3 for the *FLC* locus in WT and *emb1579* mutants. The H3K27me3 ChIP-seq peaks at the *FLC* locus in WT (red) and *emb1579* (blue) and the gene structures examined by ChIP are shown in the top, middle, and bottom rows, respectively. The purple triangles indicate specific H3K27me3 peaks in WT. The underlying numerical data are available in S1 Data. (**F**) SDS-PAGE analysis of recombinant mCherry and mCherry-MSI4. The original pictures are available in S1 Raw Images. (**G**) Visualization of MSI4 and EMB1579 in vitro under conditions that cause phase separation of EMB1579 (F-buffer: 25 mM Hepes [pH 8.0], 100 mM KCl, 100 mg/ml PEG 3350). mCherry, 12 μM; mCherry-MSI4, 20 nM; EMB1579, 2.5 μM. The partition coefficient values (*P*) were measured from 120 EMB1579 condensates and 120 MSI4 condensates. Data are presented as mean ± s.e.m. Bar = 10 μm. The underlying numerical data are available in S1 Data. (**H**) Condensation of RFP-MSI4 from *Arabidopsis* root total extract. EMB1579 was incubated with total protein extracted from roots of *Arabidopsis* expressing 35S:: RFP-MSI4. The incubation was carried out in P buffer (25 mM Tris-HCl [pH 8.0], 100 mM KCl, 2 mM DTT, 100 mg/ml PEG 3350). The lower panel shows the total protein extract without EMB1579. Bar = 2 μm. (**I**) Micrographs of *Arabidopsis* protoplasts derived from WT or *emb1579* expressing RFP-MSI4. White triangles indicate MSI4 compartments. Bars = 5 μm. (**J**) A simple model describing the function of EMB1579 in regulating the transcription of *FLC* and flowering. EMB1579 condenses the CUL4-DDB1^MSI4 complex to facilitate its interaction with CLF-PRC2 complex. This increases the level of H3K27me3 on the *FLC* locus and represses the expression of *FLC* to promote flowering. ChIP-seq, chromatin immunoprecipitation sequencing; CLF-PRC2, CURLY LEAF containing Polycomb Repressive Complex 2; CUL4, Cullin 4; DDB1, DNA Damage Binding Protein 1; EMB1579, EMBRYO DEFECTIVE 1579; *FLC*, *FLOWERING LOCUS C*; H3K27me3, trimethylation of lysine 27 of histone H3; MSI4, MULTIPLE SUPPRESSOR OF IRA 4; RFP, red fluorescent protein; WT, wild type.

indeed recruit and condense DDB1B in vitro (S9 Fig). Therefore, we speculate that EMB1579 performs its cellular functions by forming condensates that can recruit and condense CUL4-DDB1$^{MSI4}$ complex to increase their local concentration. This will facilitate the interaction of CUL4-DDB1$^{MSI4}$ complex with CLF-PRC2 complex to promote its role in establishing and/or maintaining the level of H3K27me3 on *FLC*, thereby repressing *FLC* transcription and promoting flowering (Fig 6J).

## Discussion

We here found that EMB1579 is involved in numerous physiological processes and is crucial for *Arabidopsis* growth and development. We demonstrate that EMB1579 forms liquid-like condensates in vivo and is able to undergo LLPS in vitro. We provided direct evidence that the RED repeat is important for EMB1579 to form normal liquid-like condensates and is crucial for EMB1579 to perform its physiological functions. Using floral transition as the representative physiological process, we demonstrate that EMB1579 is involved in *FLC*-mediated repression of flowering, presumably by interacting with MSI4, CUL4, and DDB1 and condensing them through the formation of liquid-like condensates. Consequently, EMB1579 might facilitate the interaction of CUL4-DDB1$^{MSI4}$ complex with CLF-PRC2 to promote its role in establishing and/or maintaining the level of H3K27me3 on *FLC*, thereby repressing the expression of *FLC* and promoting flowering in *Arabidopsis*. This work provides insights into both plant development regulatory programs and liquid-like biological systems and raises many intriguing questions for future research.

### EMB1579 compartments are highly dynamic phase-separated condensates

Consistent with the presence of IDRs in EMB1579 (Fig 2I), we found that EMB1579 undergoes LLPS in vitro (Fig 2K). EMB1579 compartments are roughly spherical and fuse with each other both in vitro and in vivo, and they undergo rapid internal rearrangement (Fig 2F–2H, Fig 2M–2O). These characteristics support the notion that EMB1579 compartments are liquid-like condensates. Strikingly, our data suggest that EMB1579 compartments are extremely dynamic, as the half-time of recovery of EMB1579 compartments was determined to be 7.4 seconds in vivo (Fig 2G) and 2.61 seconds in vitro (Fig 2N). The extraordinarily dynamic properties of EMB1579 compartments might be due to the fact that the EMB1579 protein is mostly occupied by IDRs (Fig 2I), which do not adopt stable secondary and tertiary structures and are conformationally heterogeneous and dynamic. Although EMB1579 is conserved in higher plants (S10 Fig), no full-length homologues of EMB1579 have been found in mammalian systems. Therefore, no side-by-side comparison can be performed in terms of the LLPS dynamics of EMB1579 condensates. However, EMB1579 condensates are much more dynamic than some previously reported subnuclear condensates in mammalian systems. For instance, the half-time of recovery is about 23 seconds for nucleophosmin (NPM1) [63] and 19–64 seconds for the RBP hnRNPA1 [64]. It was reported that cytoplasmic elements in stationary plant cells (e.g., cortical microtubules and the actin cytoskeleton [65,66]) are more dynamic than their counterparts in mammalian systems, and this was proposed to be an adaptation strategy for sessile plants. In this regard, it might be fair for us to speculate that the extraordinarily dynamic EMB1579 condensates enable sessile plants to rapidly alter their DNA- and RNA-based activities to control transcription and pre-mRNA splicing in response to external and internal stimuli.

### The RED repeat modulates the formation of EMB1579 condensates, and it is required for EMB1579 to perform cellular functions

More and more proteins have been shown to undergo LLPS and participate in numerous fundamental physiological processes [19,67]. For most of these proteins, however, the direct link

between the formation of condensates and their physiological functions still remains to be established. Here, we specifically identify that the RED repeat is important for the formation of normal-sized EMB1579 condensates in vitro and in vivo (Fig 5C–5J), and we demonstrate that the RED repeat is required for EMB1579 to perform cellular functions (Fig 5K, S6C–S6F Fig). Our finding that alternating basic-acidic dipeptide repeats are required for the formation of functionally normal EMB1579 condensates enhances our understanding of the principles that govern the in vivo dynamics of liquid-like bodies. Some other proteins containing RED, ER, or RD repeats also form subnuclear compartments in cells [60–62] and have been implicated in fundamental physiological processes. It is interesting to speculate whether the basic-acidic dipeptide repeats have a conserved function in LLPS of those proteins and formation of functionally competent condensates. Our study lays a foundation for further deep dissection of the mechanism of action of proteins containing RED, ER, or RD alternating basic-acidic dipeptide repeats.

## EMB1579 compartments condense different biomolecules and, specifically, they regulate the function of MSI4

The nature of the functions that are performed by nonmembranous compartments and how those functions are performed are two fundamental questions in the phase-separation field. Consistent with our results that many nuclear proteins crucial for DNA and RNA biology are pulled down by EMB1579 (Fig 3A, S1 Table), we found that global gene transcription and mRNA splicing were altered in *emb1579* mutants (Fig 3B and 3C, S2 and S3 Tables). This explains why loss of function of *EMB1579* causes pleiotropic developmental defects in *Arabidopsis* (Fig 1). Importantly, some proteins crucial for gene transcription and pre-mRNA splicing colocalize with EMB1579 condensates in vivo (Fig 4A), which allows us to speculate that EMB1579 condensates regulate gene transcription and mRNA splicing by recruiting and condensing those proteins (Fig 4C). We showed that MSI4 appears in the EMB1579 pull-down fraction (Fig 3A, S1 Table), and *emb1579* mutants have similar *FLC*-related phenotypes to *msi4* (Figs 3C, 6B, 6C and 6E). Therefore, to understand how EMB1579 performs its physiological functions, we further investigated the functional relationship between EMB1579 and MSI4 in the *FLC*-mediated repression of flowering. MSI4 interacts with CUL4-DDB1 and CLF-PRC2 [36], and we demonstrated that EMB1579 is involved in the transcription of *FLC* by interacting with CUL4-DDB1$^{MSI4}$ complex but not CLF-PRC2. Based on our results, we propose that EMB1579 functions by forming condensates that recruit and condense CUL4-DDB1$^{MSI4}$ complex to facilitate the interaction of this complex with CLF-PRC2, which in turn controls the level of H3K27me3 on *FLC* and the transcription of *FLC* to regulate flowering (Fig 6J). Condensation of the interactors of PRC2 through subcompartmentalization may provide a way to regulate the function of PRC2 in establishing and/or maintaining the level of H3K27me3 on chromatin. In particular, condensation of the scaffold protein MSI4 may facilitate the assembly of functional PRC2 complexes, although it was shown that MSI4 is not the core subunit of EMF complex [35]. Considering that CUL4 interacts with *FLC* chromatin in a MSI4-dependent manner [36], the condensation of CUL4-DDB1$^{MSI4}$ complex might facilitate the targeting and/or recruitment of PRC2 to chromatin. Our study enriches our understanding of transcriptional regulation via PRC2, a major regulator of specific gene expression patterns and developmental programs in different organisms. Interestingly, it was shown that some PcG complex proteins can undergo LLPS and form liquid-like condensates in the nucleus [68,69]. It will be interesting to investigate how EMB1579 condensates function coordinately with liquid-like condensates formed by PcG complex proteins, if plant PcG complex proteins can also undergo LLPS. Nonetheless, our study significantly enhances our understanding of how LLPS systems perform cellular functions.

These data together allow us to propose that EMB1579 controls *Arabidopsis* growth and development by forming nonmembranous compartments that recruit and condense important biomolecules to drive complex biochemical reactions. Two important questions for future research are how EMB1579 condensates selectively condense different players and how EMB1579 condensates sense intrinsic and environmental cues to adjust their dynamic properties to meet different physiological demands. It was proposed that EMB1579 is involved in nuclear calcium signaling during the salt response, since it contains an EF-hand motif [44]. It will be interesting to investigate whether EMB1579 condensates can sense nuclear calcium signaling through the EF-hand to adjust their phase-separation dynamics. In addition, MAP190, the tobacco homologue of EMB1579, can interact with actin filaments and microtubules [43], and cytoskeletal systems are able to phase separate [70,71]. In the future, it will be important to determine how LLPS of EMB1579 links the nuclear skeleton to gene transcription and mRNA splicing.

## Materials and methods

### Plant materials and growth conditions

The T-DNA insertion lines CS16026 and Salk_007142 were designated as *emb1579-1* and *emb1579-3*, respectively. The mutant *mis4* (Sail_1167_E05) has been described previously [36]. The genotyping of T-DNA insertion mutants was performed with the primers listed in S4 Table. *Arabidopsis* Columbia-0 (Col-0) ecotype was used as WT, and plants were grown in soil or media under a 16-hour-light/8-hour-dark photoperiod at 22°C.

### Construction of plasmids

The promoter of *EMB1579*, a 2,278-bp DNA fragment upstream of the initiation ATG of *EMB1579*, was amplified with primers proEMB1579-SalI-F and proEMB1579-BamHI-R (S4 Table) from *Arabidopsis* genomic DNA and subsequently moved into pCAMBIA1301 to generate pCAMBIA1301-proEMB1579. The genomic sequence of *EMB1579* (*gEMB1579*, 6,686 bp) was amplified with primers gEMB1579-SmaI-F and gEMB1579-SacI-R (S4 Table) using *Arabidopsis* genomic DNA as the template and cloned into pEASY-Blunt to generate pEASY-Blunt-gEMB1579. Three *TGFP* and *gEMB1579* were sequentially moved into pCAMBIA1301-proEMB1579 to generate pCAMBIA1301-proEMB1579::gEMB1579-TGFP. To delete the RED repeat flanking the sequence from 1461 to 1694 in *gEMB1579*, a PCR fragment was amplified with primers ΔRED-F and ΔRED-R (S4 Table) using the *pEASY-Blunt-gEMB1579* plasmid as the template and moved into pCAMBIA1301-proEMB1579-TGFP digested with *Sma*I and *Sac*I to generate pCAMBIA1301-proEMB1579::gEMB1579ΔRED-TGFP.

To investigate the tissue expression pattern of *EMB1579*, the region containing the promoter and the first two exons of *EMB1579* was amplified with primers proEMB1579-SalI-F and GUS-BamHI-R (S4 Table) using *Arabidopsis* genomic DNA as the template. The amplified PCR product was cloned into pBI101, which contains the GUS coding region, to generate pBI101-EMB1579pro::GUS. To observe the fluorescence heterogeneity of EMB1579ΔRED-TGFP, pCAMBIA1301-35S::GFP-NLS-NOS was constructed as a control. The promoter of 35S was amplified with primer pair 35S-HindIII-F/35S-PstI-R (S4 Table), using *pFGC5941* as the template, and subsequently cloned into pCAMBIA1301 to generate pCAMBIA1301-35S. GFP-NLS and NOS were amplified with primer pairs GFP-NLS-SacI-F/GFP-NLS-EcoRI-R and NOS-EcoRI-F/NOS-BstXI-R (S4 Table) using *pEZS-NL* and *pBINPLUS-35S::U2B″-RFP-NOS* as the template, respectively. These two fragments were cloned into pCAMBIA1301-35S to generate pCAMBIA1301-35S::GFP-NLS-NOS.

To determine whether proteins of interest colocalize with EMB1579 compartments, *mRFP* fusion constructs of the proteins were generated. *Arabidopsis* seedling cDNA was used as the template to amplify the protein coding sequences. The *U2B″* coding sequence was amplified with primers U2B"-SalI-F and U2B"-KpnI-R (S4 Table), and *mRFP* with a GGGG linker was amplified with primers RFP-BamHI-linker-F and RFP-HindIII-R (S4 Table). *U2B″* and *mRFP* were sequentially moved into pBINPLUS-35S-NOS to generate pBINPLUS-35S::U2B″-RFP-NOS. The *mRFP* fusion constructs of *U1-70K*, *HYL1*, *SC35*, *DDB1B*, *RZ-1C*, and *MSI4* were generated using the homologous recombination strategy. The full-length cDNAs of *U1-70K*, *HYL1*, *SC35*, *DDB1B*, *RZ-1C*, and *MSI4* were amplified with primer pairs U1-70K-AscI-F/U1-70K-linker-R, HYL1-AscI-F/HYL1-linker-R, SC35-AscI-F/SC35-linker-R, DDB1B-AscI-F/DDB1B-RFP-R, RFP-RZ-1C-SwaI-F/RFP-RZ-1C-PacI-R, and RFP-MSI4-SwaI-F/RFP-MSI4-PacI-R (S4 Table), respectively; and the corresponding mRFP fragments were amplified with primer pairs U1-70K-linker-RFP-F/linker-RFP-PacI-R, HYL1-linker-RFP-F/linker-RFP-PacI-R, SC35-linker-RFP-F/linker-RFP-PacI-R, DDB1B-linker-RFP-F/linker-RFP-PacI-R, and RFP-AscI-F/RFP-SwaI-R for *RZ-1C* and *MSI4* fragments (S4 Table), respectively. The amplified cDNA and the corresponding *mRFP* fragment were incubated with pFGC5941 linearized with *Asc*I and *Pac*I to perform recombination reactions to generate pFGC5941-U1-70K-RFP, pFGC5941-HYL1-RFP, pFGC5941-SC35-RFP, pFGC5941-DDB1B-RFP, pFGC5941-RFP-RZ-1C, and pFGC5941-RFP-MSI4 using a TIANGEN kit according to the manufacturer's instructions.

To observe EMB1579 colocalization with MSI4, the EMB1579 coding region was amplified with primer pair gEMB1579-SmaI-F/gEMB1579-SacI-R (S4 Table) using *Arabidopsis* seedlings' cDNA as the template and cloned into pCAMBIA1301-35S to generate pCAMBIA1301-35S::EMB1579. GFP and NOS were amplified with primer pairs GFP-NLS-SacI-F/GFP-EcoRI-R and NOS-EcoRI-F/NOS-BstXI-R (S4 Table) using *pEZS-NL* and *pBINPLUS-35S::U2B″-RFP-NOS* as the template, respectively. These two fragments were cloned into pCAMBIA1301-35S::EMB1579 to generate pCAMBIA1301-35S::EMB1579-GFP-NOS.

To determine whether proteins of interest can physically interact with EMB1579 or EMB1579ΔRED, the coding regions of 23 genes were amplified with the following primer pairs: EMB1579/EMB1579ΔRED—EMB1579-KpnI-F/EMB1579-SalI-R, MSI4—MSI4-KpnI-F/MSI4-SalI-R, DDB1B—DDB1B-KpnI-F/DDB1B-SalI-R, CUL4—CUL4-KpnI-F/CUL4-SalI-R, FIE—FIE-KpnI-F/FIE-SalI-R, CLF—CLF-KpnI-F/CLF-SalI-R, CDKC;2—CDKC;2-KpnI-F/CDKC;2-SalI-R, SKIP—SKIP-KpnI-F/SKIP-SalI-R, UAP56A—UAP56A-KpnI-F/UAP56A-SalI-R, SC35—SC35-KpnI-F/SC35-SalI-R, U1-70K—U1-70K-KpnI-F/U1-70K-SalI-R, RZ-1C—RZ-1C-KpnI-F/RZ-1C-SalI-R, EBP1—EBP1-KpnI-F/EBP1-SalI-R, RBP47C—RBP47C-KpnI-F/RBP47C-SalI-R, LIF2—LIF2-KpnI-F/LIF2-SalI-R, RBP45A—RBP45A-KpnI-F/RBP45A-SalI-R, BTF3—BTF3-KpnI-F/BTF3-SalI-R, MED8—MED8-KpnI-F/MED8-SalI-R, RAD23C—RAD23C-KpnI-F/RAD23C-SalI-R, RAD51D—RAD51D-KpnI-F/ RAD51D-SalI-R, NRP1—NRP1-KpnI-F/NRP1-SalI-R, CHR28—CHR28-KpnI-F/CHR28-SalI-R (S4 Table). The CDSs of the genes were incubated with pCAMBIA1300-NLuc or pCAMBIA1300-CLuc linearized with *Kpn*I and *Sal*I to perform recombination reactions to generate pCAMBIA1300-EMB1579-NLuc, pCAMBIA1300-EMB1579ΔRED-NLuc, pCAMBIA1300-CLuc-MSI4, pCAMBIA1300-CLuc-DDB1B, pCAMBIA1300-CLuc-CUL4, pCAMBIA1300-CLuc-FIE, pCAMBIA1300-CLuc-CLF, pCAMBIA1300-CLuc-CDKC;2, pCAMBIA1300-CLuc-SKIP, pCAMBIA1300-CLuc-UAP56A, pCAMBIA1300-CLuc-SC35, pCAMBIA1300-CLuc-U1-70K, pCAMBIA1300-CLuc-RZ-1C, pCAMBIA1300-CLuc-EBP1, pCAMBIA1300-CLuc-RBP47C, pCAMBIA1300-CLuc-LIF2, pCAMBIA1300-CLuc-RBP45A, pCAMBIA1300-CLuc-BTF3, pCAMBIA1300-CLuc-MED8, pCAMBIA1300-CLuc-RAD23C, pCAMBIA1300-CLuc-RAD51D, pCAMBIA1300-CLuc-NRP1, and pCAMBIA1300-CLuc-CHR28.

To determine the intracellular localization of MSI4 in WT and *emb1579 Arabidopsis* protoplasts, the CDS of *MSI4* was amplified from cDNA of *Arabidopsis* seedlings with primer pair MSI4-KpnI-F2/MSI4-XbaI-R (S4 Table), and *mRFP* was amplified with primer pair RFP-SalI-F/RFP-KpnI-R (S4 Table) for the fusion with *MSI4*. *MSI4* and *mRFP* were subsequently moved into pEZS-NL to generate pEZS-NL-mRFP-MSI4.

To generate an *FLC*-RNAi construct, the sequence containing the second and third exons of *FLC* was amplified with primers FLC-RNAi-XbaI-AscI-F containing *Xba*I and *Asc*I sites and FLC-RNAi-BamHI-SwaI-R containing *Bam*HI and *Swa*I sites (S4 Table), using *Arabidopsis* seedlings' cDNA as the template. The amplified inverted-repeat PCR fragment was subsequently digested with *Asc*I/*Swa*I or *Xba*I/*Bam*HI to generate two complementary RNAi fragments, which were moved into pFGC5941 to generate the pFGC5941-*FLC*-RNAi construct.

To generate a plasmid expressing recombinant EMB1579 protein, the full-length *EMB1579 CDS* was amplified from *Arabidopsis* seedling cDNA with primers EMB1579-EcoRI-F and EMB1579-SalI-R2 (S4 Table) and subsequently moved into pFastBac1 to generate pFastBac1-EMB1579. To generate EMB1579ΔRED, PCR was performed with primers ΔRED-F and ΔRED-R (S4 Table) using *pFastBac1-EMB1579* as the template to generate pFastBac1-EMB1579ΔRED.

To generate the recombinant GFP protein as the control for determining the concentration of EMB1579 in the nucleus, *GFP* was amplified with primer pair GFP-SalI-F/GFP-XbaI-R (S4 Table) and moved into pColdI digested with *Sal*I and *Xba*I to generate pColdI-GFP.

To determine whether EMB1579 compartments can recruit DDB1B and MSI4, *mCherry* was initially amplified with primer pair mCherry-NdeI-F/mCherry-linker-KpnI-R (S4 Table) and moved into pColdI digested with *Nde*I and *Kpn*I to generate pColdI-mCherry. *MSI4* was amplified with primer pair MSI4-SalI-F/MSI4-XbaI-R2 (S4 Table) using *pEZS-NL-RFP-MSI4* plasmid as the template and subsequently moved into pColdI-mCherry digested with *Sal*I/ *Xba*I to generate pColdI-mCherry-MSI4. To generate the mCherry-DDB1B fusion construct, *mCherry* and *DDB1B* were amplified with primer pairs mCherry-XhoI-F/mCherry-linker-EcoRI-R and DDB1B-EcoRI-F/DDB1B-SalI-R2 (S4 Table) using *pColdI-mCherry* and *pFGC5941-DDB1B-RFP* plasmid as the template, respectively. After digestion with *Xho*I/*Eco*RI and *Eco*RI/*Sal*I, respectively, *mCherry* and *DDB1B* were subsequently moved into pColdI linearized with *Xho*I and *Sal*I to generate pColdI-mCherry-DDB1B.

To determine whether EMB1579 (N500, N500ΔRED, M1, M2, and C296) directly interacts with other proteins, *EMB1579* was amplified with primer pair EMB1579-SfiI-F/EMB1579-SfiI-R (S4 Table) using *pFastBac1-EMB1579* as the template and moved into pGBKT7 digested with *Sfi*I to generate pGBKT7-EMB1579. To map the binding domain that interacts with different proteins in EMB1579, four consecutive fragments—N500, M1, M2, and C296—were amplified with primer pairs EMB1579-EcoRI-F/N500-SalI-R, M1-EcoRI-F/M1-SalI-R, M2-EcoRI-F/M2-SalI-R, and C296-EcoRI-F/C296-SalI-R (S4 Table) using *pFastBac1-EMB1579* as the template and cloned into pGBKT7 linearized with *Eco*RI and *Sal*I to generate pGBKT7-N500, pGBKT7-M1, pGBKT7-M2, and pGBKT7-C296, respectively. To determine whether the RED repeat contributes the interaction of N500 with MSI4, *N500ΔRED* was amplified with EMB1579-EcoRI-F/N500-SalI-R (S4 Table) using *pFastBac1-EMB1579ΔRED* as the template and cloned into pGBKT7 linearized with *Eco*RI and *Sal*I to generate pGBKT7-N500ΔRED.

The full-length cDNAs of *DDB1B*, *MSI4*, *CUL4*, *FIE*, and *CLF* were amplified using cDNA from *Arabidopsis* seedlings as the template with primer pairs DDB1B-EcoRI-F/DDB1B-XhoI-R, MSI4-NdeI-F/MSI4-XhoI-R, CUL4-EcoRI-F/CUL4-BamHI-R, FIE-BamHI-F/FIE-SacI-R, and CLF-NdeI-F/CLF-EcoRI-R (S4 Table), respectively. The CDSs of *DDB1B*, *MSI4*, *CUL4*, *FIE*, and *CLF* were cloned into pGADT7 linearized with *Eco*RI/*Xho*I, *Nde*I/*Xho*I, *Eco*RI/

*Bam*HI, *Bam*HI/*Sac*I, and *Nde*I/*Eco*RI to generate pGADT7-DDB1B, pGADT7-MSI4, pGADT7-CUL4, pGADT7-FIE, and pGADT7-CLF, respectively.

### *Agrobacterium*-mediated *Arabidopsis* transformation

The plasmids were introduced into *Agrobacterium tumefaciens* strain GV3101. *Agrobacterium*-mediated *Arabidopsis* transformation was performed by the floral dip method [72]. *pBI101-EMB1579pro::GUS* was transformed into WT *Arabidopsis* plants; *pCAMBIA1301-proEMB1579::gEMB1579-TGFP* and *pCAMBIA1301-proEMB1579::gEMB1579ΔRED-TGFP* were transformed into *emb1579* plants to generate pCAMBIA1301-proEMB1579::gEMB1579-TGFP; *emb1579* and pCAMBIA1301-proEMB1579::gEMB1579ΔRED-TGFP; *emb1579* plants, respectively. *pBINPLUS-35S::U2B″-RFP-NOS*, *pFGC5941-U1-70K-RFP*, *pFGC5941-HYL1-RFP*, *pFGC5941-SC35-RFP*, *pFGC5941-DDB1B-RFP*, and *pFGC5941-RFP-RZ-1C* were transformed into pCAMBIA1301-proEMB1579::gEMB1579-TGFP; *emb1579* plants; *pFGC5941-RFP-MSI4* was introduced into WT plants and *pFGC5941-FLC-RNAi* was transformed into *emb1579* plants.

### Transient expression in *Arabidopsis* mesophyll protoplasts

*Arabidopsis* mesophyll protoplasts were isolated from WT or *emb1579 Arabidopsis* plants according to the published method [73]. The plasmid *pEZS-NL-RFP-MSI4* was introduced into protoplasts derived from WT or *emb1579* plants via the PEG-calcium-mediated transformation method [73]. The transformed protoplasts were observed under an Olympus FV1200MPE laser scanning confocal microscope 8–12 hours after transformation. mRFP was excited with Argon 561-nm laser line.

### Firefly split luciferase complementation imaging assay

The *A. tumefaciens* strain GV3101 was cultured in LB liquid medium with 50 μg/ml kanamycin and 50 μg/ml rifampicin at 28°C overnight. Bacterial cells were pelleted under 4,000$g$ for 10 minutes at room temperature and subsequently resuspended with 10 mM MES (pH 5.6), 10 mM MgCl$_2$, 100 μM acetosyringone to a final concentration of OD$_{600}$ = 0.8. After 2–4 hours' incubation at room temperature, the strains containing different plasmids were mixed equivoluminally and infiltrated into 3-week-old *N. benthamiana* leaves. After 60 hours of infiltration, the leaves were sprayed with 1 mM D-luciferin and the signal was detected with a NightShade LB 985 plant imaging system.

### Transient expression in *N. benthamiana*

*A. tumefaciens* strain GV3101 was transformed with *pCAMBIA1301-35S::EMB1579-GFP-NOS*, *pFGC5941-RFP-MSI4*, and p19. The three transformants were mixed equivoluminally and infiltrated into 3-week-old *N. benthamiana* leaves. The procedures and buffers were prepared as previously described [74]. After 60 hours of infiltration, the leaf epidermal cells were observed under an Olympus FV1200MPE laser scanning confocal microscope using 488-nm excitation for GFP and 561-nm excitation for RFP.

### Particle bombardment

Transient expression of GFP-NLS in the leaves of 3- to 4-week-old WT *Arabidopsis* plants was achieved using a biolistic DNA delivery system. The procedure of bombardment of leaves using *35S::GFP-NLS-NOS* plasmid was performed according to the published method [74]. After bombardment, the leaves were placed on MS medium with 0.8% agar and incubated at

23˚C for 48 hours in the dark. Leaves were observed at 488-nm excitation for GFP fluorescence.

## Yeast two-hybrid assay

The bait and prey plasmids were cotransformed into *Saccharomyces cerevisiae* reporter strain Gold competent cells according to the manufacturer's instructions (Clontech). After growing on SD/-Leu/-Trp medium at 30˚C for 2–3 days, the transformants were selected on SD/-His/-Leu/-Trp/-Ade medium at 30˚C for 3 days to detection interactions. White yeast that grows well on SD/-His/-Leu/-Trp/-Ade medium indicates a positive interaction.

## RT-PCR analysis

Total RNA was extracted from 2-week-old *Arabidopsis* seedlings. After treatment with DNase I (M0303S, NEB), first-strand cDNA was synthesized with 10 μg total RNA using M-MLV Reverse Transcriptase (M170A, Promega) and oligo $d(T)_{18}$ (3806, TaKaRa) primer or random primer (for mRNA splicing detection). The truncated *EMB1579* transcripts in *emb1579-1* and *emb1579-3* were detected by semiquantitative RT-PCR using three primer pairs as follows: for *emb1579-1*, EMB1579-$C_F$-RT/EMB1579-$C_R$-RT, EMB1579-$D_F$-RT/EMB1579-$D_R$-RT, and EMB1579-$E_F$-RT/EMB1579-$D_R$-RT; for *emb1579-3*, EMB1579-$A_F$-RT/EMB1579-$A_R$-RT, EMB1579-$B_F$-RT/EMB1579-$B_R$-RT, and EMB1579-$C_F$-RT/EMB1579-$C_R$-RT (S4 Table). The full-length transcripts of *EMB1579* were amplified with primer pair EMB1579-qRT-F2/EMB1579-qRT-R2 (S4 Table). *eIF4A* was amplified with primer pair eIF4A-F1/eIF4A-R1 (S4 Table) as the internal control. The transcript level of *EMB1579* was also detected with qRT-PCR using primer pair EMB1579-qRT-F1/EMB1579-qRT-R1 (S4 Table). *eIF4A* was amplified with primer pair eIF4A-F2/eIF4A-R2 (S4 Table) as the internal control. The relative mRNA expression levels in samples were calculated according to the $2^{-\triangle\triangle Ct}$ method [75]. qRT-PCR was performed with primers FLC intron1 SF and FLC intron1 SR (S4 Table) to detect the expression level of *FLC* in WT and *emb1579* plants. To detect the splicing status (spliced or unspliced) of the first intron of *FLC*, PCR products were amplified with primer pairs FLC intron1 UF/FLC intron1 UR and FLC intron1 SF/FLC intron1 SR (S4 Table) using cDNA from WT and *emb1579* seedlings by random primed reverse transcription [11]. To determine the splicing status (spliced or unspliced) of the third intron of *ICK2*, primer pairs ICK2-SF/ICK2-SR and ICK2-UF/ICK2-UR (S4 Table) were used, respectively. For *CYCD2;1*, primer pairs CYCD2;1-SF/CYCD2;1-SR and CYCD2;1-UF/CYCD2;1-UR (S4 Table) were used to measure the levels of abnormal *CYCD2;1* transcripts, which lack the second and third exons, and normal *CYCD2;1* transcripts, which contain the second and third exons. The splicing efficiency was determined by normalizing the level of spliced transcripts to the level of unspliced transcripts for each splicing defect as described previously [11,76]. qRT-PCR was conducted using 2×RealStar Green Power Mixture (with ROX II) in an Applied Biosystems 7500 Fast Real-Time PCR System. All primers used in the qRT-PCR for the validation of RNA-seq data are listed in S5 Table. All data were obtained from three biological replicates.

## Preparation of nuclear proteins and western blot analysis

Nuclear proteins were extracted from 2-week-old *Arabidopsis* seedlings according to a previously published method [77]. The nuclear proteins were separated on SDS-PAGE gels and transferred to 0.45-μm nitrocellulose membranes (Amersham). The membranes were blocked with 5% skim milk powder (BD) in 50 mM Tris-buffered saline (TBS) containing 0.1% Tween 20 (TBS-T) at room temperature for 1 hour and subsequently incubated with anti-H3 (Abcam, ab1791) (diluted at 1:3,000 in TBS-T with 5% skim milk powder), anti-H3K27me3 (Millipore,

07–449) (diluted at 1:500) [78,79], or anti-GFP (EASYBIO, BE2002) (diluted at 1:1,000) at room temperature for 1 hour. The membranes were washed three times with TBS-T, then incubated with goat anti-rabbit IgG-HRP antibody (EASYBIO, BE0101) (diluted at 1:10,000) for 1 hour at room temperature. Finally, the membranes were washed three times with TBS-T, and the protein bands were detected using a chemiluminescent system (Thermo, 34075).

## Determination of the concentration of EMB1579 in the nucleus

Briefly, 2-week-old proEMB1579::gEMB1579-TGFP; *emb1579 Arabidopsis* seedlings (approximately 3 g) were ground in liquid nitrogen. Subsequently, a certain amount (V1 mL) of extraction buffer (20 mM Tris-HCl [pH 7.4], 25% glycerol, 20 mM KCl, 2 mM EDTA, 2.5 mM MgCl$_2$, 250 mM sucrose, 5 mM DTT), supplemented with complete EDTA-free Protease Inhibitor Cocktail, was added into the ground mixture. After centrifugation at 10,000$g$ for 10 minutes at 4˚C, the supernatant at V2 mL was collected. Next, the nuclei were isolated and total nuclear proteins were extracted as described previously [77]. The amount of EMB1579-TGFP (ng) in the total nuclear protein was determined by western blot analysis probed with anti-GFP antibody, using a known amount of recombinant GFP protein (1–10 ng) as the control. To calculate the molar concentration of EMB1579-TGFP in the nucleus, we initially assumed that the volume of cells is equal to (V1 − V2) mL and the nucleus occupies roughly 2.6% of the whole cell volume, as reported previously [80]. The amount of EMB1579-TGFP (ng) was finally divided by the volume of nuclei ([V1 − V2]*2.6%) and the molecular weight of EMB1579-TGFP (260 kDa) to yield the molar concentration of EMB1579-TGFP (nM) in the nucleus, as described previously [81].

## GUS staining

The seedlings were immersed in the GUS staining solution (0.5 mg/ml X-glucuronide in 100 mM sodium phosphate [pH 7.0], 0.5 mM ferricyanide, 0.05 mM ferrocyanide, 1 mM EDTA, and 0.1% Triton X-100) for 14 hours at 37˚C. GUS-stained materials were cleared of chlorophyll in 70% (v/v) ethanol. To perform GUS staining in seeds, the siliques were dissected to expose seeds. The seeds were initially incubated with GUS staining solution for 20 minutes under vacuum infiltration, followed by an additional incubation for 4 hours at 37˚C. Finally, the embryos were cleared by HCG solution (eight parts chloroacetaldehyde: three parts water: one part glycerol). Stained materials were photographed under an Olympus IX83 microscope.

## Imaging of *Arabidopsis* root and embryo cells

To image *Arabidopsis* root cells and embryonic cells, propidium iodide (PI) staining was performed. Briefly, *Arabidopsis* roots were incubated with PI at 20 μg/ml for about 5 minutes. After rinsing with ddH$_2$O, they were mounted in ddH$_2$O for visualization. The staining of embryos with PI was performed according to a previously published method [45]. Briefly, embryos were isolated from ovules and subsequently incubated with PI solution (50 μg/ml; dissolved in 9% glucose and 5% glycerol) for about 10 minutes before observation. *Arabidopsis* roots and embryos were observed under an Olympus FV1200MPE laser scanning confocal microscope with the excitation wavelength set at 543 nm for PI staining. To visualize *Arabidopsis* embryos at earlier stages, the ovules were dissected and cleared in Hoyer's solution [82] from 2 hours to overnight depending on developmental stage and were subsequently observed under an Olympus BX53 microscope equipped with differential interference contrast (DIC) optics.

## Complementation of *emb1579* and visualization of the intracellular localization and dynamics of EMB1579

To visualize the intracellular localization of EMB1579-TGFP or EMB1579ΔRED-TGFP, roots from pCAMBIA1301-proEMB1579::gEMB1579-TGFP; *emb1579* seedlings or pCAM-BIA1301-proEMB1579::gEMB1579ΔRED-TGFP; *emb1579* seedlings that were vertically cultured on petri dishes for 2–3 days were observed under a fluorescence laser scanning confocal microscope equipped with a ×100 oil objective (numerical aperture of 1.4) at 488-nm excitation. The Z-series images were collected with the step size set at 0.5 μm. The nuclei were stained with Hoechst 33342 (30 μg/ml), which was excited with a 405-nm laser.

To capture the dynamics of EMB1579-TGFP during the cell cycle, the sample was visualized by a spinning disk confocal microscope, and the time-series Z-stack images were collected at 2-minute intervals with the step size set at 1 μm. The collected images were processed with ImageJ software. To reveal the internal dynamics of EMB1579 bodies, FRAP was carried out with an Olympus FV1200MPE laser scanning confocal microscope. Two scan images were captured before bleaching and a single body was bleached using 100% power with 488-nm and 10% power with 405-nm laser lines for 5 seconds. Time-lapse images were collected at 4-second intervals. The gray values of single bodies were analyzed with ImageJ software against the elapsed time to calculate the rate of fluorescence recovery. The fluorescence intensity of pre-bleaching was normalized to 100%. Tracking of body fusion events was performed in root meristem cells. The difference between EMB1579-TGFP and EMB1579ΔRED-TGFP in forming subnuclear compartments was assessed by measuring the coefficient of variation of GFP fluorescence in the nucleoplasm with ImageJ software as described previously [83].

## Protein production

To generate recombinant EMB1579 and EMB1579ΔRED proteins, the constructs *pFastBac1-EMB1579* and *pFastBac1-EMB1579ΔRED* were transformed into DH10Bac competent cells and screened by blue/white selection with 100 μg/ml Bluo-gal, 40 μg/ml IPTG, 50 μg/ml kanamycin, 7 μg/ml gentamycin, and 10 μg/ml tetracycline in LB agar plates. The recombinant bacmids extracted from the white colonies were transfected into Sf9 insect cells with Cellfectin II Reagent. After two generations of virus propagation, the virus was inoculated into Sf9 insect cells for large-scale protein production. The transfected cells were collected at 1,000$g$ for 10 minutes under 4˚C, suspended in binding buffer (25 mM Tris-HCl [pH 8.0], 250 mM KCl, 5 mM imidazole, 1 mM PMSF) and sonicated on ice using an ultrasonic cell disruptor (27 times for 2 seconds at 40% power). The sonicate was centrifuged at 14,000$g$ for 40 minutes under 4˚C, and the supernatant was incubated with Ni-NTA beads (Novagen). After extensive washing with washing buffer (25 mM Tris-HCl [pH 8.0], 250 mM KCl, 10 mM imidazole), the target protein was eluted with elution buffer (25 mM Tris-HCl [pH 8.0], 250 mM KCl, 250 mM imidazole). EMB1579 and EMB1579ΔRED proteins were further purified with SuperDex 200 Increase columns (GE Healthcare) pre-equilibrated with buffer X (25 mM Tris-HCl [pH 8.0], 250 mM KCl, 2 mM DTT).

To generate recombinant GFP, mCherry, mCherry-MSI4, and mCherry-DDB1B proteins, the plasmids *pColdI-GFP*, *pColdI-mCherry*, *pColdI-mCherry-MSI4*, or *pColdI-mCherry-DDB1B* were transformed into *Escherichia coli* BL21 (DE3) strain. The cells were cultured for 4 hours at 37˚C until the OD$_{600}$ reached 0.8, and the expression of recombinant proteins was induced by the addition of 0.4 mM IPTG at 15˚C for 19 hours. The bacterial cells were collected by centrifugation at 5,000$g$ for 10 minutes under 4˚C, resuspended in binding buffer (25 mM Tris-HCl [pH 8.0], 250 mM KCl, 5 mM imidazole, 1 mM PMSF), and sonicated on ice using an ultrasonic cell disruptor (41 times for 7 seconds at 30% power). The sonicate was

clarified by centrifugation under 15,000$g$ for 45 minutes at 4˚C, and the supernatant was incubated with Ni-NTA beads. After extensive washes with washing buffer (25 mM Tris-HCl [pH 8.0], 250 mM KCl, 10 mM imidazole), the recombinant proteins were eluted with elution buffer (25 mM Tris-HCl [pH 8.0], 250 mM KCl, 250 mM imidazole). GFP, mCherry, mCherry-MSI4, and mCherry-DDB1B proteins were dialyzed against 5 mM Tris-HCl (pH 8.0).

### Preparation of fluorescently labeled EMB1579 and its variants

To allow direct visualization of phase separation by fluorescence light microscopy, purified recombinant EMB1579 and EMB1579ΔRED proteins dialyzed against buffer B (25 mM Hepes [pH 8.0], 250 mM KCl) were subjected to fluorescent labeling with Oregon Green (see below). After the proteins were allowed to undergo phase separation in F-buffer (25 mM Hepes [pH 8.0], 100 mM KCl, 100 mg/ml PEG 3350) for 1 hour at room temperature, an 8-fold molar ratio of Oregon Green was added and the mixture was further incubated in F-buffer for 90 minutes at room temperature. The reaction mixtures were centrifuged under 15,000$g$ for 30 minutes at room temperature, and the pellet was washed with F-buffer twice to remove the free Oregon Green and subsequently resuspended in buffer E (25 mM Hepes [pH 8.0], 500 mM KCl). The suspension was clarified under 15,000$g$ at 4˚C for 30 minutes, and the supernatant was used for the subsequent phase-separation experiments.

### Phase transition of EMB1579 and its variants in vitro

EMB1579 and its variants were preclarified under 14,000$g$ for 20 minutes at 4˚C before the reaction. To allow the phase separation to occur in vitro, a range of EMB1579 concentrations from 0.01 μM to 1 μM and a range of EMB1579ΔRED concentrations from 0.05 μM to 1 μM were incubated in P buffer (25 mM Tris-HCl [pH 8.0], 100–2,000 mM KCl, 2 mM DTT). The phase separation of EMB1579 and its variants was directly visualized by either DIC optics or fluorescent light microscopy. To monitor the dynamic phase-separation process, the protein mixture in P buffer with 100 mg/ml PEG 3350 was injected into a flow chamber with the coverslip pretreated with Poly-L-lysine. Time-lapse images were collected at 10-second intervals.

To reveal the dynamic exchange of bodies formed by Oregon Green-EMB1579 and Oregon Green-EMB1579ΔRED, FRAP experiments were performed under an Olympus FV1200MPE laser scanning confocal microscope. For the whole-body FRAP, the bleaching was performed by intense illumination with the 488-nm laser set to 100% and the 405-nm laser set to 40% for 5 seconds. For the half-bleach, the setting was 100% power with 488-nm and 405-nm laser lines for 0.08 seconds. Time-lapse images were collected at 3-second intervals for the whole-body FRAP and 0.2-second intervals for half-FRAP. The rate of fluorescence recovery was measured and calculated with ImageJ using the same method as described above for EMB1579-TGFP FRAP in vivo.

To determine the saturation curve of EMB1579 and EMB1579ΔRED, EMB1579 and EMB1579ΔRED were preclarified under 14,000$g$ for 20 minutes at 4˚C. EMB1579 ranging from 200 to 500 nM and EMB1579ΔRED ranging from 200 to 1,000 nM were incubated in P buffer (25 mM Tris-HCl [pH 8.0], 100–3,000 mM KCl, 2 mM DTT) and centrifuged at 600$g$, 20˚C for about 3 hours [84]. The supernatant was collected and the protein concentration in the supernatant was measured using the Bradford assay and A280.

### Visualization of the recruitment and condensation of proteins by compartments formed by EMB1579 in vitro

EMB1579, mCherry-MSI4, mCherry-DDB1B, and mCherry were preclarified under 14,000$g$ for 20 minutes at 4˚C. To determine whether MSI4 and DDB1B can be recruited into

EMB1579 compartments, Oregon Green–labeled EMB1579 (2.5 μM) was incubated with mCherry-DDB1B (500 nM) or mCherry-MSI4 (20 nM) or mCherry (12 μM) in F-buffer for 5 minutes at room temperature in the dark. mCherry-DDB1B, mCherry-MSI4, or mCherry was used as the control under the same conditions. The mixtures were injected into a flow chamber with the coverslip pretreated with Poly-L-lysine and observed under an Olympus FV1200MPE laser scanning confocal microscope with excitation wavelengths set at 488 nm for Oregon Green and 561 nm for mCherry. The partition coefficient was defined as the ratio of the protein in condensates versus that in dilute phase, which was calculated according to the published method [85].

To determine whether EMB1579 condensates can condense MSI4 from *Arabidopsis* total protein extract, 2-week-old *Arabidopsis* roots expressing *pFGC5941-RFP-MSI4* were collected and ground in buffer (50 mM Tris-HCl [pH 8.0], 2 M KCl, 4 mM DTT, complete EDTA-free Protease Inhibitor Cocktail) on ice. The extract was centrifuged under 14,000*g* at 4˚C for 20 minutes and the supernatant (containing the protein) was subsequently separated into two equal parts. One part was dialyzed with EMB1579 in buffer P with PEG 3350 for phase separation and the other part was added into the same volume of buffer X and dialyzed in buffer P with PEG 3350 as a control. After 1-hour dialysis, the mixed proteins were injected into a flow chamber with the coverslip pretreated with Poly-L-lysine and observed under an Olympus FV1200MPE laser scanning confocal microscope with excitation wavelength set at 561 nm for RFP.

## Pull-down experiments and mass spectrometry analysis

Two-week-old roots of WT *Arabidopsis* seedlings were collected and soaked in protein extraction buffer (50 mM Tris-HCl [pH 8.0], 50 mM KCl, 2 mM DTT) for about 1 hour. The protein extraction buffer was discarded and the samples were ground on ice with the addition of complete EDTA-free Protease Inhibitor Cocktail. The ground samples were subsequently subjected to centrifugation under 14,000*g* at 4˚C for 20 minutes. The collected supernatant was dialyzed into P buffer (25 mM Tris-HCl [pH 8.0], 50 mM KCl, 2 mM DTT), and recombinant EMB1579-6×His protein bound to Ni-NTA beads was subsequently added into the supernatant. Ni-NTA beads were used as the control. After incubation for 2 hours on ice, the Ni-NTA beads were briefly washed twice with buffer P. The protein samples were separated by SDS-PAGE and detected by silver staining. The bands of interest were cut out and the proteins were identified by mass spectrometry (OrbiTrap Fusion LUMOS, Thermo).

## RNA-seq

Total RNA was isolated from 7-day-old WT and *emb1579-3 Arabidopsis* seedlings with TRIzol. Seedling cDNA libraries for WT and *emb1579-3* were constructed using RNA-seq for paired-end transcriptome sequencing, which was conducted by Shanghai Majorbio Biopharm Technology (Shanghai, China). Poly(A) mRNA was isolated using Sera-mag Magnetic Oligo (dT) Beads and fragmented into short pieces. Double-strand cDNA was obtained with random hexamers and reverse transcriptase. RNA-seq libraries were constructed and sequenced using Illumina HiSeq 4000 in Shanghai Majorbio Bio-pharm Biotechnology (Shanghai, China). The RNA-seq reads were aligned to the TAIR10 *Arabidopsis* Genome using TopHat v2.0 [86].

## ChIP-seq

Chromatin was extracted from 7-day-old *Arabidopsis* seedlings of WT and *emb1579-3* mutants. Anti-H3K27me3 antibodies were used for ChIP. After sonication to break the DNA into pieces, the DNA fragments isolated by ChIP were used to construct DNA pools with a

NEBNext Ultra II DNA Library Prep Kit for Illumina following the manufacturer's instructions. After six cycles of PCR, PCR products were confirmed to be in the size range 200–500 bp by DNA agarose gel electrophoresis. The 200- to 500-bp DNA bands were collected and further used for PCR amplification. The PCR products were purified with magnetic beads and used for sequencing, which was carried on an Illumina HiSeq with sequencing by synthesis in Shanghai Romics Biotechnology (Shanghai, China).

## Supporting information

**S1 Fig. Expression pattern of *EMB1579*.** The tissue expression pattern of *EMB1579* was revealed by monitoring the activity of GUS in transgenic plants harboring the fusion construct *EMB1579pro*-GUS. (**A**) Expression of *EMB1579* in embryos at different stages. Bar = 50 μm. (**B**) Expression of *EMB1579* in seedlings. Bars = 2 mm. (**C**) Expression of *EMB1579* in root. Bar = 100 μm. (**D**) Expression of *EMB1579* in flowers. The bar is 1.5 mm (left panel), 100 μm (middle panel), and 2 mm (right panel). (**E**) Expression of *EMB1579* in a leaf branch. Bar = 1 mm. *EMB1579*, EMBRYO DEFECTIVE 1579.
(TIF)

**S2 Fig. Complementation of *emb1579* mutants.** To complement *emb1579* mutants, the construct *pCAMBIA1301-proEMB1579:: gEMB1579-TGFP* was transformed into *emb1579-1* and *emb1579-3* mutants. (**A**, **F**) RT-PCR analysis of the level of *EMB1579* transcripts in WT, *emb1579*, and two complementation lines. The original pictures are available in S1 Raw Images. (**B**, **G**) Images of 7-day-old seedlings of WT, *emb1579*, and two complementation lines. Bars = 1.5 cm. (**C**, **H**) Quantification of primary root length of 7-day-old seedlings of WT, *emb1579*, and complementation lines. Data are presented as mean ± s.e.m. ***$P < 0.001$ by Student *t* test. Numerical data underlying the panels are available in S1 Data. (**D**) Images of 6-week-old *Arabidopsis* plants of WT, *emb1579-1*, and a complementation line. Bar = 2 cm. (**E**, **J**) Quantification of the number of rosette leaves at bolting in WT, *emb1579*, and complementation lines. Data are presented as mean ± s.e.m. ***$P < 0.001$ by Student *t* test. Numerical data underlying these panels are available in S1 Data. (**I**) Images of 7-week-old *Arabidopsis* plants of WT, *emb1579-3*, and a complementation line. Bar = 2 cm. *emb1579*, embryo defective 1579; ND, no significant difference; RT-PCR, reverse transcription PCR; WT, wild type.
(TIF)

**S3 Fig. EMB1579 condensates are distinct from CB and HYL1 bodies.** (**A**) Micrograph of *Arabidopsis* root cells expressing *proEMB1579::gEMB1579-TGFP* and *35S::U2B''-RFP*. Bar = 5 μm. (**B**) Micrograph of *Arabidopsis* root cells expressing *proEMB1579::gEMB1579-TGFP* and *35S::HYL1-RFP*. Bar = 5 μm. EMB1579, EMBRYO DEFECTIVE 1579.
(TIF)

**S4 Fig. Determination of the concentration of EMB1579 in the nucleus.** (**A**) Western blot analysis of nuclear proteins from proEMB1579::gEMB1579-TGFP; *emb1579*. The western blot was probed with anti-GFP antibody. The original pictures are available in S1 Raw Images. (**B**) Western blot analysis of recombinant GFP protein (1–10 ng), which was used as a loading control to quantify EMB1579-TGFP. The western blot was probed with anti-GFP antibody. The original pictures are available in S1 Raw Images. (**C**) Quantification of the concentration of EMB1579-TGFP in the nucleus. The abundance of EMB1579-TGFP protein was defined as the ratio of the amount of EMB1579-TGFP versus the amount of total protein. The value is presented as mean ± SD. EMB1579, EMBRYO DEFECTIVE 1579; GFP, green fluorescent protein; TGFP, tandem copies of enhanced GFP.
(TIF)

**S5 Fig. Reducing the expression of *FLC* alleviates the later flowering phenotype in *emb1579* mutants.** (**A**) RNAi-mediated knock-down of *FLC* in *emb1579* plants. Relative expression of *FLC* was determined by qRT-PCR analysis. Data are presented as mean ± s.e.m, *n* = 3. *FLC*-RNAi plants are *emb1579* plants expressing *pFGC5941-FLC-RNAi*. Numerical data underlying this panel are available in S1 Data. (**B**) Images of 6-week-old *Arabidopsis* plants. Bar = 2 cm. *emb1579*, embryo defective 1579; *FLC*, *FLOWERING LOCUS C*; qRT-PCR, quantitative reverse transcription PCR; RNAi, RNA interference.
(TIF)

**S6 Fig. EMB1579ΔRED fails to function properly in vivo.** (**A**) qRT-PCR analysis to determine the relative level of *EMB1579* transcripts in WT, *emb1579*, and the complementation plants. EMB1579-TGFP, expression of EMB1579-TGFP under control of the EMB1579 promoter in *emb1579* mutants; EMB1579ΔRED-TGFP, expression of EMB1579ΔRED-TGFP under control of the EMB1579 promoter in *emb1579* mutants. Data are presented as mean ± s. e.m, *n* = 3. Numerical data underlying this figure are available in S1 Data. (**B**) Western blot analysis to determine the relative amount of EMB1579-TGFP and EMB1579ΔRED-TGFP in the nucleus. Total nuclear proteins from *Arabidopsis* seedlings were probed with anti-GFP antibody. H3 protein (detected with an anti-H3 antibody) was used as the loading control. The original pictures are available in S1 Raw Images. (**C**) Images of 7-day-old *Arabidopsis* seedlings growing on plates. Bar = 0.5 cm. (**D**) Quantification of primary root length of 7-day-old seedlings in WT, *emb1579*, and its complementation lines. Data are presented as mean ± s.e.m. ***P < 0.001 by Student *t* test. Numerical data underlying this panel are available in S1 Data. (**E**) Images of 6-week-old *Arabidopsis* plants growing in pots. Bar = 2 cm. (**F**) Quantification of the number of rosette leaves at bolting in WT, *emb1579*, and the complementation lines. Data are presented as mean ± s.e.m. ***P < 0.001 by Student *t* test. Numerical data underlying this panel are available in S1 Data. EMB1579, EMBRYO DEFECTIVE 1579; ND, no significant difference; qRT-PCR, quantitative reverse transcription PCR; TGFP, tandem copies of enhanced green fluorescent protein; WT, wild type.
(TIF)

**S7 Fig. Deletion of the RED repeat does not disrupt the interaction of EMB1579 with its functionally relevant interactors.** Interactions were detected by the firefly split luciferase complementation imaging assay. In total, 20 EMB1579-interacting proteins were tested for their interaction with EMB1579ΔRED. EMB1579, EMBRYO DEFECTIVE 1579.
(TIF)

**S8 Fig. EMB159 interacts with MSI4, DDB1B, and CUL4 but not CLF and FIE.** (**A**) The firefly split luciferase complementation imaging assay was used to determine the interactions of EMB1579 with CUL4, FIE, and CLF. (**B**) Yeast two-hybrid analysis of the interactions of EMB1579 with MSI, CLF, FIE, CUL4, and DDB1B. (**C**) Mapping the binding region of MSI4 in EMB1579. The left panel shows schematic diagrams of EMB1579 and its truncations. The red box represents the RED repeat. The right panel shows yeast two-hybrid analysis of the interaction of the truncated EMB1579 proteins with MSI4. (**D**) Yeast two-hybrid analysis was performed to detect the interactions between N500 and MSI4, CLF, FIE, CUL4, or DDB1B. CLF, CURLY LEAF; CUL4, Cullin 4; DDB1, DNA Damage Binding Protein 1; EMB1579, EMBRYO DEFECTIVE 1579; FIE, FERTILIZATION INDEPENDENT ENDOSPERM; MSI4, MULTIPLE SUPPRESSOR OF IRA 4.
(TIF)

**S9 Fig. EMB1579 condensates can recruit and condense DDB1B in vitro.** (**A**) SDS-PAGE analysis of recombinant mCherry-DDB1B. The original pictures are available in S1 Raw Images. (**B**) Visualization of DDB1B and EMB1579 in vitro under conditions that cause phase separation of EMB1579 (F-buffer: 25 mM Hepes [pH 8.0], 100 mM KCl, 100 mg/ml PEG 3350). mCherry, 12 μM; mCherry-DDB1B, 0.5 μM; EMB1579, 2.5 μM. The partition coefficient values were measured from 112 EMB1579 condensates and 112 DDB1 condensates. Data are presented as mean ± s.e.m. Bar = 10 μm. The underlying numerical data are available in S1 Data. DDB1, DNA Damage Binding Protein 1; EMB1579, EMBRYO DEFECTIVE 1579.
(TIF)

**S10 Fig. Phylogenetic analysis of EMB1579 and its homologues in the plant kingdom.** The phylogenetic tree of EMB1579 and its homologues was constructed with MEGA5.0 software. The accession numbers of EMB1579 and its homologues can be found either in GenBank or at the website http://congenie.org/ as follows: *Arabidopsis thaliana*, NP_178414; *Aegilops tauschii*, XP_020167627; *Amborella trichopoda*, XP_006827314; *Brachypodium distachyon*, XP_003563745; *Brassica rapa*, XP_009114177; *Camelia sativa*, XP_010425097; *Capsella rubella*, XP_006290321; *Capsicum annuum*, XP_016572460; *Chlamydomonas reinhardtii*, XP_001701130; *Citrus sinensis*, XP_006483121; *Cucumis sativus*, XP_011655281; *Cynara cardunculus*, KVH95716; *Erythranthe guttata*, EYU30242; *Eucalyptus grandis*, XP_010066890; *Eutrema salsugineum*, XP_006395736; *Fragaria vesca*, XP_004297287; *Glycine max*, XP_006573124; *Gossypium arboretum*, KHG15037; *Malus domestica*, XP_008339877; *Morus notabilis*, XP_010108695; *Nicotiana tabacum*, XP_016481812; *Oryza sativa*, XP_015643243; *Physcomitrella patens*, XP_001777802; *Picea taeda*, PITA_000003061; *Prunus mume*, XP_016647712; *Selaginella moellendorffii*, XP_002980580; *Solanum pennellii*, XP_015073405; *Tarenaya hassleriana*, XP_010554305; *Theobroma cacao*, EOY01867; *Triticum urartu*, EMS67387; *Vitis vinifera*, XP_010651850; *Zea mays*, XP_008649139. The gray boxes represent conserved RED repeats. EMB1579, EMBRYO DEFECTIVE 1579.
(TIF)

**S1 Movie. Time-lapse images of EMB1579-TGFP during the cell cycle in *Arabidopsis* root cells.** Time-series images of EMB1579-TGFP compartments were collected every 2 minutes, then compressed into a movie with a display rate of 3 frames per second. The white arrows indicate the disappearance of EMB1579 at the beginning of mitosis whereas the black arrow indicates the appearance of EMB1579 after mitosis. Bar = 5 μm. EMB1579, EMBRYO DEFECTIVE 1579; TGFP, tandem copies of enhanced green fluorescent protein.
(AVI)

**S2 Movie. FRAP analysis of EMB1579 compartments in *Arabidopsis* root cells.** Time-lapse images were captured every 4 seconds, and compressed into a movie with a display rate of 3 frames per second. The white box indicates the bleached EMB1579 compartment. Bar = 5 μm. EMB1579, EMBRYO DEFECTIVE 1579; FRAP, fluorescence recovery after photobleaching.
(AVI)

**S3 Movie. Time-lapse images of EMB1579-TGFP indicating condensate fusion in vivo.** Time-series images of EMB1579-TGFP condensates were collected every 4 seconds and then compressed into a movie with a display rate of 3 frames per second. The fusion event is indicated by white arrows. Bar = 5 μm. EMB1579, EMBRYO DEFECTIVE 1579; TGFP, tandem copies of enhanced green fluorescent protein.
(AVI)

**S4 Movie. Time-lapse images indicating the fusion of EMB1579 condensates in vitro.** Time-lapse images of EMB1579 compartments were collected every 10 seconds and then compressed into a movie with a display rate of 10 frames per second. The fusion events are indicated by different-colored arrows. Bar = 5 μm. EMB1579, EMBRYO DEFECTIVE 1579. (AVI)

**S5 Movie. Time-lapse images indicating the recovery of fluorescence of an EMB1579 condensate after half of it was bleached in vitro.** Half-bleaching was performed with an in vitro–formed EMB1579 condensate. Time-lapse images were captured every 0.2 seconds and then compressed into a movie with a display rate of 15 frames per second. Bar = 1 μm. EMB1579, EMBRYO DEFECTIVE 1579. (AVI)

**S1 Table. Mass spectrometry identification of proteins sedimented with EMB1579.** EMB1579, EMBRYO DEFECTIVE 1579. (XLS)

**S2 Table. Differentially expressed genes between WT and *emb1579*.** *emb1579*, embryo defective 1579; WT, wild type. (XLS)

**S3 Table. Differentially spliced genes between WT and *emb1579* based on RNA-seq.** *emb1579*, embryo defective 1579; RNA-seq, RNA sequencing; WT, wild type. (XLS)

**S4 Table. Primers used in this study.** (DOC)

**S5 Table. qRT-PCR validation primers.** qRT-PCR, quantitative reverse transcription PCR. (DOC)

**S1 Data. Individual sheets for the underlying numerical data for Figs 1–6 and figures in supporting information.** (XLS)

**S1 Raw Images. Unprocessed images of all gels and blots in the paper.** (PDF)

## Acknowledgments

We thank NASC for the sequence-indexed T-DNA insertion lines and the members of the Huang Lab for helpful discussion. We also thank the Protein Chemistry Facility at the Center for Biomedical Analysis of Tsinghua University for mass spectrometry and sample analysis.

## Author Contributions

**Conceptualization:** Shanjin Huang.

**Data curation:** Yiling Zhang, Zhankun Li, Naizhi Chen, Yao Huang.

**Formal analysis:** Naizhi Chen.

**Funding acquisition:** Shanjin Huang.

**Investigation:** Yiling Zhang, Zhankun Li, Naizhi Chen, Yao Huang.

**Methodology:** Naizhi Chen.

**Project administration:** Shanjin Huang.

**Resources:** Shanjin Huang.

**Software:** Yiling Zhang, Zhankun Li, Naizhi Chen.

**Supervision:** Shanjin Huang.

**Validation:** Yiling Zhang, Zhankun Li, Naizhi Chen, Yao Huang.

**Visualization:** Yiling Zhang, Zhankun Li, Naizhi Chen, Yao Huang.

**Writing – original draft:** Yiling Zhang, Shanjin Huang.

**Writing – review & editing:** Shanjin Huang.

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
