## [Editor Report · Decision Letter 0]

28 Oct 2019

Dear Dr Huang, 

Thank you for submitting your manuscript entitled "Phase Separation of Arabidopsis EMB1579 Controls Transcription, mRNA Splicing and Development" for consideration as a Research Article by PLOS Biology.

Your manuscript has been evaluated by the PLOS Biology editorial staff, as well as by an academic editor with relevant expertise. I am writing to let you know that your manuscript has been sent out for external peer review.

Please note, however, that the outcome of our discussion of your manuscript is that we have some reservations as to whether the data fully support the stated conclusions. We will need to be persuaded by the reviewers that the paper has the potential after revision to offer the significant strength of advance that we require for publication in order to pursue it further for PLOS Biology

Please re-submit your manuscript within two working days, i.e. by Oct 30 2019 11:59PM.

Kind regards,

Lauren A Richardson, Ph.D

Senior Editor

PLOS Biology

---

## [Decision Letter · Decision Letter 1]

21 Nov 2019

Dear Dr Huang,

Thank you very much for submitting your manuscript "Phase Separation of Arabidopsis EMB1579 Controls Transcription, mRNA Splicing and Development" for consideration as a Research Article at PLOS Biology. Your manuscript has been evaluated by the PLOS Biology editors, an Academic Editor with relevant expertise, and by independent reviewers.

As you will read, the reviewers appreciated many aspects of your study. However, they also raise some concerns that will need to be addressed in a revision. Most critically, two of the reviewers believe that the work defining the function of this phase separation and its impact on plant phenotypes requires further support. We remain interested in your study and we would be willing to consider resubmission of a comprehensively revised version that thoroughly addresses all the reviewers' comments. We cannot make any decision about publication until we have seen the revised manuscript and your response to the reviewers' comments. Your revised manuscript would be sent for further evaluation by the reviewers.

We appreciate that these requests represent a great deal of extra work, and we are willing to relax our standard revision time to allow you six months to revise your manuscript. Please email us (plosbiology@plos.org) to discuss this if you have any questions or concerns, or think that you would need longer than this. At this stage, your manuscript remains formally under active consideration at our journal; please notify us by email if you do not wish to submit a revision and instead wish to pursue publication elsewhere, so that we may end consideration of the manuscript at PLOS Biology.

Your revisions should address the specific points made by each reviewer. Please submit a file detailing your responses to the editorial requests and a point-by-point response to all of the reviewers' comments that indicates the changes you have made to the manuscript. In addition to a clean copy of the manuscript, please upload a 'track-changes' version of your manuscript that specifies the edits made. This should be uploaded as a "Related" file type. You should also cite any additional relevant literature that has been published since the original submission and mention any additional citations in your response. 

Before you revise your manuscript, please review the following PLOS policy and formatting requirements checklist PDF: http://journals.plos.org/plosbiology/s/file?id=9411/plos-biology-formatting-checklist.pdf. It is helpful if you format your revision according to our requirements - should your paper subsequently be accepted, this will save time at the acceptance stage.

Please note that as a condition of publication PLOS' data policy (http://journals.plos.org/plosbiology/s/data-availability) requires that you make available all data used to draw the conclusions arrived at in your manuscript. If you have not already done so, you must include any data used in your manuscript either in appropriate repositories, within the body of the manuscript, or as supporting information (N.B. this includes any numerical values that were used to generate graphs, histograms etc.). For an example see here: http://www.plosbiology.org/article/info%3Adoi%2F10.1371%2Fjournal.pbio.1001908#s5.

For manuscripts submitted on or after 1st July 2019, we require the original, uncropped and minimally adjusted images supporting all blot and gel results reported in an article's figures or Supporting Information files. We will require these files before a manuscript can be accepted so please prepare them now, if you have not already uploaded them. Please carefully read our guidelines for how to prepare and upload this data: https://journals.plos.org/plosbiology/s/figures#loc-blot-and-gel-reporting-requirements.

Upon resubmission, the editors will assess your revision and if the editors and Academic Editor feel that the revised manuscript remains appropriate for the journal, we will send the manuscript for re-review. We aim to consult the same Academic Editor and reviewers for revised manuscripts but may consult others if needed.

If you still intend to submit a revised version of your manuscript, please go to https://www.editorialmanager.com/pbiology/ and log in as an Author. Click the link labelled 'Submissions Needing Revision' where you will find your submission record. 

Sincerely,

Lauren A Richardson, Ph.D

Senior Editor

PLOS Biology

Reviews

Reviewer #1: 

The manuscript entitled, “Phase Separation of Arabidopsis EMB1579 Controls Transcription, mRNA Splicing and Development” shows the EMB1579 protein affecting global gene transcription and mRNA splicing. It is quite clear about the phase separation of RMB1579 and its dynamics both in vitro and in vivo. However, there are quite a lot of overstatements on the cellular functions of EMB1579. Additionally, the lack of direct evidence on the EMB1579-MSI4 interaction impact to FLC epigenetic state is another big concern.

Therefore, I suggest declining this paper for publishing in plos biology, but with strong encouragement to resubmit the manuscript after re-arranging the content and addressing all the following major concerns:

1) It’s not right to claim that phase separation of EMB1579 condenses PRC2, because what they showed are MSI4 and DDB1A, which are actually not PRC2 subunits, only been shown interacting with PRC2 complex.

2) It is obvious that EMB1579 functions in flower transition, however, the claim on regulating the H3K27me3 level on FLC through maintaining the PRC2 complex phase transition is a great jump. To claim this, the author should show 1) EMB1579 direct interaction with PRC2 complex proteins, 2) in vivo evidence that the binding of PRC2 subunit proteins on FLC was compromised in emb1579 mutant.

3) It is very interesting that the capability of EMB1579△RED to form droplets were compromised, but the EMB1579△RED droplets are even more dynamic than EMB1579 droplets in vitro. How to explain this interesting phenomenon? 

4) Although EMB1579 itself could form phase separation, it does not necessarily mean the droplets observed is initiated by EMB1579, it could be several proteins with the property of forming phase separation co-condense, and the RED domain responsible for the multi-valent interaction. Thus, EMB1579△RED lost the ability of forming phase separation in vivo could due to the lack of the interaction with these proteins, and the biological function defects in EMB1579△RED may also be explained by the RED domain as the interacting domain with other transcription/splicing complex proteins. Thus, it is not right to claim that “we uncover a direct link between the LLPS property and their physiological functions”.

5) The authors should specify how they justify “both transcription and mRNA splicing defects in emb1579 mutants suppress the later flowering phenotype in emb1579 mutants”, because only expression level of FLC was shown in fig S4.

Minor points:

1) It is very interesting that the homolog of EMB1579, MAP190 could interacts with actin filaments and microtubules. EMB1579 could form phase separation in vivo, the actin and microtubules are also shown to form phase separation1,2, this could be a very interesting area to further link the phase separation of nucleus skeleton with transcription/splicing.

1 Weirich, K. L. et al. Liquid behavior of cross-linked actin bundles. Proceedings of the National Academy of Sciences 114, 2131, doi:10.1073/pnas.1616133114 (2017).

2 Guharoy, M., Szabo, B., Martos, S. C., Kosol, S. & Tompa, P. Intrinsic Structural Disorder in Cytoskeletal Proteins. Cytoskeleton 70, 550-571, doi:10.1002/cm.21118 (2013).

Reviewer #2: 

In this manuscript, Zhang et al. delve into functions of a plant-specific protein, EMB1579. They find that EMB1579 deletion mutants exhibit global changes in transcription and mRNA splicing. These phenotypes correlate with organismal defects in seedling growth and flowering. The authors find that EMB1579 forms punctate structures in cells that exhibit some liquid-like behaviors, such as high internal mobility and fusion. Cellular EMB1579 compartments co-localize with transcription and splicing factors, including PRC2 complex proteins MSI4 and DDB1B. These proteins also appear to directly interact in Arabidopsis. The authors show that EMB1579 protein phase separates to form distinct condensates in vitro. The in vitro EMB1579 condensates also concentrate MSI4 and DDB1B, suggesting that the cellular and in vitro compartments have certain similarities. The authors further identify a region in EMB1579, RED repeats, which alters aspects of EMB1579 phase separation in cells and in vitro. RED repeat deletion mutant does not fully rescue the defects of EMB1579 deletion mutant, indicating that this region is indeed important for the function of EMB1579. In summary, this study describes a novel function of EMB1579 in transcription and mRNA splicing, which correspond to interesting plant phenotypes. In addition, the fact that the cellular tendency of this protein to localize to nuclear foci correlates with its ability to phase separate in vitro is interesting. However, as I outlined in detail below, many of the conclusions around the functional significance of EMB1579 phase separation or the link between EMB1579’s molecular function and plant phenotypes are not strongly supported. Moreover, the intro and discussion are currently plant-specific and could be expanded to be accessible to a broader PLOS Biology audience. Apart from these general comments, several points stood out to this reviewer as critical to address.

Major comments:

• The intro and discussion will benefit from linking the findings from the current study to what is already known out there. Please consider expanding on the general implications of the findings.

o PCG complex proteins have been shown to phase separate before, which is not at all mentioned in the manuscript 

o what are some analogous nuclear bodies or proteins in non-plants and what do we know about them (such as other PRC proteins that phase separate); what other analogous RED repeat containing proteins exist and what do they do; specify which proteins or nuclear compartments have been shown to use this mechanism to phase separate…etc.

• This study doesn’t specifically test the role of IDRs in phase separation, so please downplay the link between IDRs and phase separation in the text (currently it implies that the high degree of disorder in EMB1579 may contribute to its phase separation)

• The conclusions rely on in part findings in Figure 4, in which the authors look at changes in transcription and splicing in ‘phase separating conditions’. To support the conclusions, it will be crucial to show that EMB1579 phase separation in plant extracts is not disrupted during pull-down and that the assay isolates specifically the phase separated compartments. Without this information we can’t conclude that these interactions specifically occur in phase separated compartments. Consequently, I also do not agree that the model in Fig 4 is well supported.

• Some of the key findings rely on the differences between WT and RED deletion mutant. However, I could not find information on whether the expression and localization of these proteins are comparable in plants. It is essential to show this with quantification

o in Fig 4 does the RED mutant express at the same level? From the staining, it looks like expression of the mutant may be lower. Please show by western blot and quantification. 

• Relating to the in vitro phase separation assays, it would be important to include whether phase separation occurs at near physiological protein and salt concentrations. i.e. measure approximate concentration of the protein in plant cells.

• In Fig 2K, there are number of in vitro EMB1579 condensates that are touching but not fusing in both the still images and in the montage. The shapes of the condensates are generally not spherical either. This is not what you expect from dynamic liquid droplets and contradictory to the rapid fusion seen in cells, please comment on this and show additional quantification to support sphericity or fusion dynamics.

• The fact that the RED mutant does phase separate to form puncta in vitro but not in cells was strange to this reviewer. Also the mutant condensates do not grow in size in vitro as concentration increases - this is surprising given the principles of phase separation and suggest that RED repeats may have specific roles in condensate growth, rather than LLPS. Please comment on this and adjust the model accordingly. 

o Related to this, the following statement is not supported by the data: "These data together suggest that the RED repeat is required for the LLPS property of EMB1579 in vitro”

• In several places the authors make the conclusion that the LLPS of EMB1579 is essential for its function: e.g. "These data together suggest that the cellular functions of EMB1579 depend on the formation of appropriate liquid-like compartments.” The authors showed that RED repeats alter EMB1579 phase separation in vitro and EMB1579 mutant plants have defects that can be partly rescued by the mutant. With current data, it could very well be that the RED repeats are important for interaction with RNA or other proteins that are essential for EMB1579 function. Pull-down experiments to show that EMB1579 RNA and protein interactions are unaffected could be additional support for their conclusion. However, ultimately, authors would need to show that these reactions are occurring ‘better’ in the condensate in vitro or in cells in order to make this conclusion. Otherwise, please soften the conclusion.

• This reviewer did not find the trimethylation defects very convincing in support of the following conclusion: "These data together allow us to propose that EMB1579 droplets condense PRC2 to facilitate its activity in maintaining the correct level of H3K27 trimethylation on FLC to regulate the transcription of FLC and flowering (Fig 6H).” Please measure trimethylation defects with multiple methods and quantify.

• The conclusion that EMB1579 condensates function to “promoting PRC2 activity in maintaining the level of H3K27 trimethylation on FLC” is too strong – the current study shows correlative data only. 

Minor comments:

• A number of grammatical errors are found throughout the manuscript – please comb through and correct

• Figures heavily rely on the legends to make sense – more detailed labelling within the figures would significantly enhance the readability. e.g.) add arrows, label which proteins are represented in the microscope images, are gels representing WB or qPCR etc.

o Fig 1B: "distorted cell division and cell expansion" It’s difficult to see what cell division and expansion defects should be seen in the images - is it increase or decrease in cell division or both? Please include arrows. 

o Fig 1C: Not sure what kinds of seed defects are found. Please add arrows

• Conclusions in the text are written with very vague language, making it difficult to understand what the authors are trying to convey. Please be more precise. e.g.) "distorted cell division and cell expansion" - is it increase or decrease or both?; "dynamic changes during the cell cycle” – what is dynamically changing? etc. 

o Fig 2C: Please specify what changes are referred to by "dynamic changes during the cell cycle”. The only difference that is obvious to this reviewer is that GFP intensity decreases sometimes, which doesn’t come back – this doesn’t seem very ‘dynamic’.

• Fig 2D: Image does not match what is reflected in quantification in Fig 2E - I see larger bodies in root meristem zone. In 2E, is the difference significant?

• Fig 2J: The purified proteins have multiple ‘contaminating bands’ - please comment on this.

• In text corresponding to Fig 5G, the authors conclude that "EMB1579�RED droplets are even more dynamic than EMB1579 droplets”. The RED mutant also has less mobile fraction. Please comment on this.

• In discussion, “it might be fair for us to speculate that the extraordinarily dynamic EMB1579 droplets enable sessile plants to rapidly alter their DNA- and RNA-based activities to control transcription and pre-mRNA splicing in response to external and internal stimuli.” This is not fitting their conclusion that RED deletion mutant was more dynamic – ie. there’s no functional data showing link between changes in dynamics and transcription/pre-mRNA splicing control

Reviewer #3: 

Review of “Phase Separation of Arabidopsis EMB1579 Controls Transcription, mRNA Splicing and Development” by Zhang et al. 

In this manuscript, the authors describe a new mechanism linking the liquid-liquid phase separation (LLPS) property of a plant specific protein, EMB1579 and its control of multiple growth and development processes in Arabidopsis through the regulation of gene transcription and mRNA splicing.

They demonstrate that EMB1579 forms dynamic LLPS droplets both in vitro and in vivo and they identify an arg/glu/asp rich region, which is important in its phase separation and which links its LLPS and its physiological roles. They showed that EMB1579 droplets co-assembles with many chromosomal and RNA-processing proteins and its loss results in altered global gene transcription and mRNA splicing. Finally, they showed that EMB1579 droplets condense and promote the activity of PRC2 to maintain the H3K27 trimethylation on the FLC locus.

The experiments in this manuscript are appropriately designed and carried out, and the findings are significant. This work will be of great interest to the general readers of PLOS Biology as well as those interested in the biochemistry/biophysics of LLPS and it physiological role. I will recommend it for publication.

Comments:

I would like to suggest to the authors to added a figure or two showing a phase separation prediction for EMB1579 protein. A recent review from my former colleagues summarize the available predictors:

First-generation predictors of biological protein phase separation by Vernon and Forman-Kay. Curr Opin Struct Biol (2019)

Considerations and challenges in studying Liquid-Liquid Phase Separation and Biomolecular Condensates by Alberti, Gladfelter and Mittag. Cell (2019) doi: 10.1016/j.cell.2018.12.035.

Minor :

there are no page numbers in the manuscript

some image quality is fuzzy especially if you magnify (e.g. fig 3A)

PCR2 was not defined when first encounted in the text. It was done later. 

Because nucleic acids can also undergo LLPS, the following statement should be changed to reflect that fact: 

“The biophysical mechanism underlying the formation of non-membranous subnuclear compartments is liquid-liquid phase separation (LLPS), which is mediated by collective protein–protein and protein–nucleic acid interactions as well as nucleic acid-acid interactions[15, 18]”. One relevant reference is “ RNA phase transitions in repeat expansion disorders by Jain and Vale. Nature (2017) doi: 10.1038/nature22386

---

## [Decision Letter · Decision Letter 2]

26 May 2020

Dear Dr Huang,

Thank you for submitting your revised Research Article entitled "Phase Separation of Arabidopsis EMB1579 Controls Transcription, mRNA Splicing and Development" for publication in PLOS Biology. I have now obtained advice from original reviewer 2 and have discussed their comments with the Academic Editor. 

Based on the review, we will accept this manuscript only if you modify the manuscript to address all the remaining points raised by reviewer 2. Importantly, you will need to clearly distinguish correlations from causal relationships in your statements. Your revision will also need to include the determination of partition coefficient. Please also make sure to address the data and other policy-related requests noted at the end of this email.

We expect to receive your revised manuscript within one month. Your revisions should address the specific points made by reviewer 2. In addition to the remaining revisions and before we will be able to formally accept your manuscript and consider it "in press", we also need to ensure that your article conforms to our guidelines. A member of our team will be in touch shortly with a set of requests. As we can't proceed until these requirements are met, your swift response will help prevent delays to publication.

*Copyediting*

*Published Peer Review History*

*Early Version*

*Submitting Your Revision*

Sincerely,

Di Jiang, PhD

PLOS Biology

DATA POLICY:

Regardless of the method selected, please ensure that you provide the individual numerical values that underlie the summary data displayed in the following figure panels as they are essential for readers to assess your analysis and to reproduce it: Figures 1AEGI, 2EGILNO, 3CH-L, 5DEGIJK, 6CE, S2CEHJ, S5A, S6ADF. NOTE: the numerical data provided should include all replicates AND the way in which the plotted mean and errors were derived (it should not present only the mean/average values).

Reviewer remarks:

Reviewer #2: The manuscript presents beautiful cell biological and developmental characterization of EMB1579 'KO' plant mutants. These mutants show changes to global transcription and RNA splicing. The authors also identify a number of proteins that interact with EMB1579, including DDB1 and MSI4. The finding that it does not interact with some of the PRC2 complex proteins also shed more light into its functional mechanism. It is striking that RED mutant EMB1579 do not rescue the defects of EMB1579 KO plants. These findings are valuable and form a strong foundation for future studies. The result that EMB1579 phase separates into very dynamic condensates is convincing, and the requirement of RED domains to make large droplets is interesting, particularly in light of their inability to rescue KO plants. The fact that these condensates can also partition DDB1 and MSI4 recapitulates in vivo findings, and thus valuable. However, the presented data does not demonstrate that the EMB1579 phase separation facilitates EMB1579 functions or how. 

This revised manuscript has meaningful additions from the last version. In my view, the focus on EMB1579 phase separation takes away from the novelty of findings mentioned above. Importantly, the manuscript still has a number of overstatements linking EMB1579 phase separation to its function, which is a big concern. Moreover, the manuscript would greatly benefit from a significant restructuring to streamline presentation. Please find detailed comments below.

A few examples of overstatements - many appear in titles. Please comb through the manuscript for additional mistakes.

1. "Many nuclear proteins bind to EMB1579 droplets and loss of function of EMB1579 affects global transcription and mRNA splicing in Arabidopsis" � There is no validation that the method isolates interactions that occur in EMB1579 condensates. There is usually a considerable amount of protein still in the dilute phase in the in vitro assays. As such, the authors would need to show that they concentrated EMB1579 condensed phase before identifying interactors to make this claim. Relating to this, please report the partition coefficient.

2. "formation of normal-sized EMB1579 droplets is crucial for its cellular functions"; "The formation of EMB1579 droplets is crucial for its cellular functions" � The findings demonstrate that the RED domain is crucial for rescuing EMB1579 KO phenotypes, and this correlates with the RED mutant's inability to form large EMB1579 condensates. To make this claim, the authors would need to show that RED mutant is equally 'functional' as full length, only disrupted in its ability to condense, but still cannot rescue EMB1579 KO phenotypes.

3. "we demonstrate that the formation of normal EMB1579 droplets is crucial for the cellular functions of EMB1579 (Fig 5K; S6C-6F Fig)" � RED domain mutant was introduced in EMB1579 KO cells to draw this conclusion. It is equally likely that RED mutant has defects in mediating EMB1579 functions.

4. "EMB1579 compartments condense different biomolecules to regulate their functions" � The authors only showed that different proteins co-localize; regulation was not shown.

5. Please consider replacing 'droplets' with 'condensates' or the like, which are more specific and accepted term.

The introduction did not provide sufficient background information relevant to the study at hand and was too general and vague at places. Please implement a clear flow, clearly stating the knowns and unknowns to set the stage for the study and relating the subject of interest to other areas of biology (including non-plant). 

6. "the molecular mechanisms underlying the organization of the compartments, what functions they can perform, and how they perform their functions remain to be documented." � This is inaccurate. Numerous molecular mechanisms underlying condensate formation has been reported in the last decade and many functions have been demonstrated, and how they do this. This information for individual condensates may not be well understood.

7. Transcription and RNA splicing have been linked to LLPS, which seems very relevant given that EMB1579 seems to control these processes. These studies are not at all mentioned or discussed.

8. Cul4-DDB1-MSI4 is a major focus of this study. Cul4-DDB1 is also a well-characterized E3 Ub-ligase, however this is not mentioned. Is this how the complex regulates histone modifications? Is MSI an F-box protein? 

9. EMB1579 is introduced abruptly in the intro.

The flow of the results section was rough; it jumped from EMB1579 phase separation to cellular functions and back, with obvious missing pieces in linking the two concepts.

10. "embryonic developmental defects at different stages, resulting from distorted cell division and cell expansion in emb1579 embryos compared to WT (Fig 1B). Specifically, we found that the division plane of cells was mispositioned and cells were swollen in emb1579 mutants (Fig 1B)" � Please point to distorted cell division, expansion, division plane, swelling.

11. "We initially analyzed the amino acid sequence of EMB1579 using different phase separation predictors reported previously [47, 48] and found that it contains IDRs that occupy most of the protein (Fig 2I)." � The cited predictors do not look at the simple presence of IDRs. They report the likelihood of a protein to phase separate based on specific parameters. If the predictors were used, please state what was found or cite correct references for protein disorder prediction.

12. The authors state that EMB1579 phase separates at 25nM and exist in cells at around 33 nM. One would predict that EMB1579 is phase separating all the time, how could this regulate specific gene transcription and splicing? Also, the in vitro condensates recover fluorescence much faster than in cells, highlighting that there are other components of the condensates that likely play important roles (at least for the material properties).

13. "Taking these data together, we propose that EMB1579 undergoes LLPS to form liquid-like droplets that condense biomolecules crucial for chromosomal function and RNA biology to control nuclear events, such as transcription and pre-mRNA splicing (Fig 4C)." � I would like to see a model like this after all relevant data is presented.

Discussion:

14. "MSI4 regulates the flowering time by forming CUL4-DDB1MSI4 complex that interacts with a CLF-PRC2 complex to repress the expression of FLC via H3K27 methylation" � Please include speculations as to how this works? Ub-ligase function or not?

15. "we speculate that EMB1579 performs its cellular functions by forming droplets that can recruit and condense CUL4-DDB1MSI4 complex to increase their local concentration. This will facilitate the interaction of CUL4-DDB1MSI4 complex with CLF-PRC2 complex to promote its role in establishing and/or maintaining the level of H3K27me3 on FLC, thereby repressing FLC transcription and promoting flowering (Fig 6J)" � The authors showed that EMB1579 does not bind PRC2 components, how would this work?

---

## [Editor Report · Decision Letter 3]

6 Jul 2020

Dear Dr Huang,

On behalf of my colleagues and the Academic Editor, Xuemei Chen, I am pleased to inform you that we will be delighted to publish your Research Article in PLOS Biology. 

Early Version

PRESS 

Kind regards,

Alice Musson

Publishing Editor, 

PLOS Biology

on behalf of

Di Jiang, PhD,

Senior Editor

PLOS Biology